# Aerosol-fog interaction and the transition to well-mixed radiation fog

Ian Boutle[1], Jeremy Price[1], Innocent Kudzotsa[2], Harri Kokkola[2], and Sami Romakkaniemi[2]

[1]Met Office, Exeter, UK
[2]Finnish Meteorological Institute, Kuopio, Finland

*Correspondence to:* I. A. Boutle (ian.boutle@metoffice.gov.uk)

**Abstract.** We analyse the development of a radiation fog event and its gradual transition from optically-thin fog in a stable boundary layer to well-mixed optically-thick fog. Comparison of observations and a detailed large-eddy simulation demonstrate that aerosol growth and activation is the key process in determining the onset of adiabatic fog. Weak turbulence and low supersaturations lead to the growth of aerosol particles which can significantly affect the visibility, but do not significantly interact with the long-wave radiation, allowing the atmosphere to remain stable. Only when a substantial fraction of the aerosol become activated into cloud droplets can the fog interact with the radiation, becoming optically thick and well-mixed. Modifications to the parametrization of cloud droplet numbers in fog, resulting in lower and more realistic concentrations, are shown to give significant improvements to an NWP model, which initially struggled to accurately simulate the transition. Finally, consequences of this work for common aerosol activation parametrizations used in climate models are discussed, demonstrating that many schemes are reliant on an artificial minimum value when activating aerosol in fog, and adjustment of this minimum can significantly affect the sensitivity of the climate system to aerosol radiative forcing.

## 1 Introduction

Radiation fog is a challenging problem for numerical weather prediction (NWP) models. To obtain an accurate forecast, models must correctly represent the coupling between the land-surface and atmosphere which leads to the fog formation, and the coupling between different atmospheric parametrizations (cloud microphysics, turbulence, radiation) which determine the evolution of the fog. It is therefore unsurprising that the quality of NWP fog forecasts remains low. Significant advancement of both the physical process understanding and modelling is required to achieve forecasts the quality of which customers have come to expect for other high-impact weather events.

Several recent studies have focussed on large-eddy simulation (LES) of foggy events (Bergot et al., 2015; Mazoyer et al., 2017; Maronga and Bosveld, 2017) to investigate the systematic response of fog to various dynamical and physical processes. All of these studies showed how quantities such as windspeed, temperature, humidity, or land surface characteristics, influence the initiation, peak intensity and dissipation of fog, often in a fairly linear way. However, Price (2011) has shown that fog can undergo some more significant bifurcations in its behaviour. He showed that, for a site in south-east England, approximately 50% of fogs remained as optically-thin condensed water in a stable boundary layer, whilst the other 50% formed optically-

thick (i.e. zero net surface radiation), well-mixed, adiabatic fog, with the radiative cooling and turbulence generated giving the fog a distinct structure of its own. Identifying, and more importantly, being able to forecast what causes this transition would be of significant value. Stable fogs generally persist throughout the night-time, dissipating quickly due to solar heating in the morning. However adiabatic fogs have the ability to persist for days, greatly increasing their potential to cause disruption to, for example, road traffic and airports.

One possible mechanism for this non-linear response could arise from atmospheric aerosol concentrations and their interaction with the developing fog layer. Bott (1991) thoroughly studied the effect of aerosol concentrations and properties on single column model simulations of the life-cycle of radiation fog. He demonstrated that as aerosol concentrations were increased, the fog became deeper, with higher condensed water content, and was less likely to be dissipated by solar radiation. Despite the crucial importance of this conclusion for NWP forecasts, most operational models used for fog prediction do not consider variable aerosol or fog droplet number concentrations (e.g. Gultepe et al., 2006; Tudor, 2010; van der Velde et al., 2010), and therefore it may be unsurprising that they struggle to obtain sufficient accuracy.

The Local and Non-local Fog Experiment (LANFEX, Price et al., 2018) was a recent UK attempt to gather new observations of foggy events to make significant progress in their understanding and modelling. This paper focusses on the first intensive observation period (IOP1), during which a shallow, optically-thin fog transitioned slowly into a well-mixed, optically-thick fog. This paper aims to extend the work of Bott (1991), investigating the role aerosol plays in this transition, and establishing whether an NWP model is capable of reproducing the observed behaviour. We discuss the current ability of an NWP model to reproduce the observations (Section 3), investigate the mechanisms causing the transition, supplementing the observational analysis with process modelling from an LES (Section 4), and evaluate some simple improvements to the NWP model (Section 5). As we find that one of the key processes is the representation of aerosol activation within the model, we conclude (Section 6) with some interesting consequences of this work for the climate system and climate modelling.

## 2   Case and model details

IOP1 took place on the night of 24-25 November 2014, and was measured at the Met Office field site at Cardington, UK (52.1015N, 0.4159W). Widespread radiation fog formed across much of the country, and remained stable for much of the night before becoming adiabatic later. IOP1 was one of the cleanest examples of local fog development observed during LANFEX, with no evidence that advective or non-local processes were significant (this has been tested in single column model simulations with and without advective forcing; not shown). A detailed set of research-grade observations are available from the Cardington site, including a 50 m flux tower, radiosonde launches, a cloud-droplet probe flown on a tethered balloon and standard surface and sub-surface measurements (see Price et al., 2018, for more details).

The NWP model considered in this paper is the Met Office Unified Model, specifically the 1.5 km horizontal grid-length 'UKV' model. Details of this model and some discussion of its ability at forecasting fog can be found in Boutle et al. (2016), although several aspects of the model dynamics (Wood et al., 2014), turbulence (Boutle et al., 2014b) and cloud microphysics (Boutle et al., 2014a) parametrizations have been significantly upgraded. For the simulations presented, the model is initialised

from its own analysis at 12 UTC on 24 November 2014, and is free-running after this, forced only at the boundaries by data from the Met Office global model (Walters et al., 2017b). The model contains 70 vertical levels, 6 of which are below 150 m and the lowest of which are at 2.5 m for horizontal winds and 5 m for temperature and humidity. As discussed in Clark et al. (2008), the model contains a single species prognostic aerosol, which is used in the diagnosis of near-surface visibility, and

converted to cloud droplet number (Wilkinson et al., 2013; Osborne et al., 2014) for use in the radiation and microphysical parametrizations, i.e. the first and second indirect effects.

To understand the physical processes leading to the observed behaviour, we also consider simulations using UCLALES–SALSA (Tonttila et al., 2017) as a process-model. This model comprises an LES model coupled to a detailed, interactive, aerosol-cloud microphysics model, and has been used previously to investigate the role aerosols play in the development of

radiation fog (Maalick et al., 2016). Within the LES model, aerosol is partitioned into 10 bins covering the size range 3 nm to 10 $\mu$m. Water condensation onto aerosol particles is calculated by numerically solving the condensation equation at every time step and grid point. Thus we are able to explicitly simulate how radiative cooling and turbulence affect the water saturation ratio and how this affects the size of the aerosol particles. If aerosol becomes activated, i.e. its size exceeds the critical size given by Köhler theory, it is transferred into a separate sectional cloud droplet model, with bin sizes matching those of the

dry cloud condensation nuclei and diagnosed wet sizes, typically between 0.7 $\mu$m and 50 $\mu$m. Thus we do not employ any traditional parametrization of activation commonly used in other models, but instead simulate the actual supersaturation and growth of aerosol particles into droplets. Comparisons with a more detailed parcel model (Kokkola et al., 2008) demonstrate that this growth is solved with good accuracy for a range of air parcels and updraft velocities.

For the simulations presented, the LES is initialised at 17 UTC with a radiosonde profile, and forced at the surface through-

out the simulation with observed surface temperatures. This ensures that the simulation remains close to the observations throughout, avoiding the large uncertainties in the land-surface model discussed by Maronga and Bosveld (2017). In this case, the surface fluxes are very small and typically negative, therefore feedbacks from the land to atmosphere are not expected to play a great role (although this will not always be the case, particularly when fluxes are larger and positive). The domain size is 500×500 m in the horizontal with a 4 m grid-length, and 700 m in the vertical with a 1.5 m grid-length below 150 m,

above which a stretching factor of 1.05 is applied. The time-step is variable, but typically around 0.25 s after the turbulence has formed. As such, the cloud activation within turbulent updrafts is well resolved, occurring on the timescale of a few seconds. The aerosol distribution was initialised with 1000 cm$^{-3}$ number concentration of Aitken mode aerosols (mean diameter 0.05 $\mu$m), 100 cm$^{-3}$ accumulation mode aerosols (mean diameter 0.15 $\mu$m) and 2 cm$^{-3}$ coarse mode aerosols (mean diameter 1 $\mu$m), each following a lognormal distribution with standard deviation of 2. Sadly, direct observations of aerosol concentra-

tions were not available for this case, but this distribution is representative of the clean air typically found at Cardington. Several different aerosol concentrations were tested and we briefly discuss the sensitivity to this choice in the conclusions. During the simulations, water condensation on cloud droplets is explicitly calculated, and collision processes between different sized hydrometeors are also accounted for. Because radiation fogs are relatively thin, and there is no real precipitation formation, the autoconversion is turned off in the model. Instead, hydrated aerosol particles and cloud droplets are allowed to sediment onto

the surface. Full details can be found in Tonttila et al. (2017) and references within.

## 3 NWP model simulations

The observations (Fig. 1a) showed a drop in visibility to ≈1 km as haze and patchy fog formed just before 18 UTC, followed by the continuous onset of thicker fog just after 20 UTC as the visibility dropped to near 100 m. The fog persisted for approximately 12 hours throughout the night, clearing at around 08 UTC the following morning.

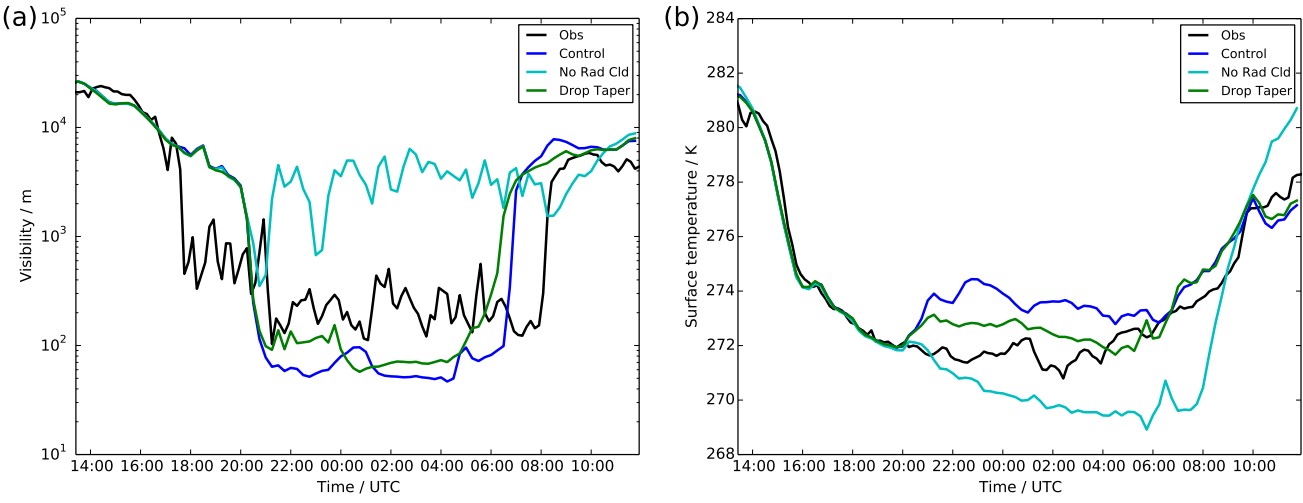

**Figure 1.** Time series of observations (black) and UKV experiments: control (blue), radiatively inactive cloud (cyan) and modified droplet number (green), showing (a) visibility, and (b) grass surface temperature.

The control UKV shows a very good simulation of the pre-fog conditions, tracing the observed surface temperature (Fig. 1b) almost perfectly until the onset of fog at 20 UTC. The model then shows a sharp drop in median visibility, down to <100 m, at approximately the same time as the observations show a drop in visibility to similar values. The model also produced non-zero probabilities of visibility <1 km from 18 UTC, therefore it appears that the onset of fog, particularly the timing, is reasonably well reproduced in this model. However, after the onset of fog, there are noticeable differences between the model

and observations. The model shows a 2 K increase in surface temperature immediately after the onset of fog, something which is not apparent in reality. It also appears that the fog is too thick in the model, with visibilities close to 60 m, compared with 200 m in the observations. The dissipation of fog also occurs about 1–1.5 hours too early in the model, although this is due to a bank of mid-level cloud arriving at the location too early in the model. The cloud causes an increase in downwelling long-wave radiation, which directly heats the surface and fog layer, causing its dissipation. In reality, it is difficult to determine whether it

was the presence of this cloud, or onset of downwelling short-wave radiation after sunrise (08 UTC) which caused dissipation, as both occurred at approximately the same time.

    Figure 2 shows the screen temperature and surface sensible heat flux from the UKV simulation. The sharp rise in surface temperature after the onset of fog leads to a positive sensible heat flux being formed throughout the night. Although modest in size (10 Wm$^{-2}$), it is in clear contrast to the observations which remain near zero throughout, and drives a warming of the

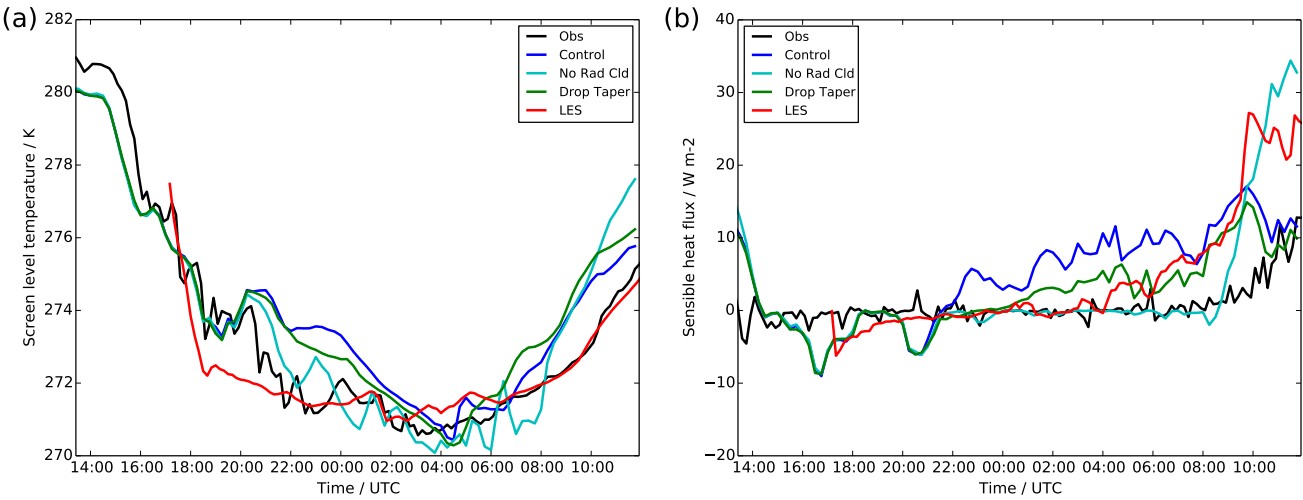

**Figure 2.** Time series of observations (black), LES (red) and UKV experiments: control (blue), radiatively inactive cloud (cyan) and modified droplet number (green), showing (a) screen (1.5 m) temperature, and (b) surface sensible heat flux.

screen temperature which therefore remains warmer than the observations. Figure 3 shows the effect this has throughout the

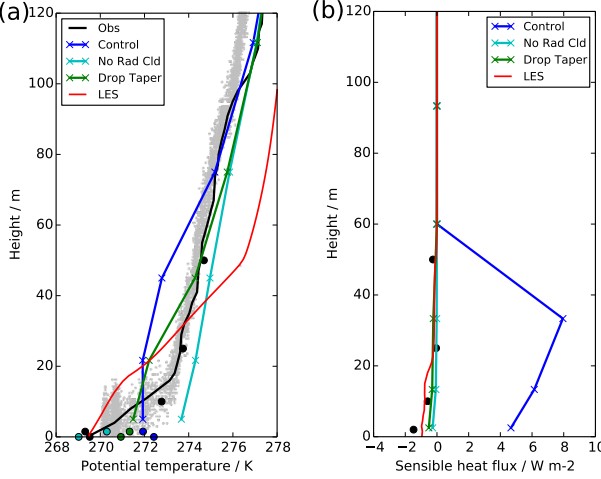

**Figure 3.** Profiles at 22.30 UTC showing observations (mast: black circles, radiosonde: black line, tethered balloon: grey dots), LES (red) and UKV experiments: control (blue), radiatively inactive cloud (cyan) and modified droplet number (green), showing (a) potential temperature, and (b) sensible heat flux. Model profiles show model-level data (crosses) and diagnosed screen and surface level temperature (filled circles).

boundary layer. The positive heat flux exists through the lowest 50 m of the atmosphere before dropping to zero at what is likely to be the boundary layer top. The observations by contrast show a near zero flux throughout the depth of the boundary

layer. The consequence of this for the temperature profiles is that reality maintains a stable boundary layer, whilst the model has a well-mixed temperature profile indicative of an unstable boundary layer.

To understand what is going wrong in the model, it is useful to consider the typical evolution of a real-world radiation fog event, such as this one. Long-wave radiative cooling of the land surface gives rise to a stable boundary layer profile, and a
thin fog will form within this profile when the relative humidity reaches 100%. Initially this fog will only interact weakly with the environment, allowing the surface and air temperature to continue to cool and the fog to thicken. It will take several hours for the fog to start to develop turbulent behaviour of its own. At this point, long-wave radiative cooling from the fog top will generate negatively buoyant air parcels, which will descend throughout the fog layer and generate turbulence. Long-wave emission from the fog layer will also act to heat the surface, which is now 'insulated' from the clear sky by the fog layer, and
this in turn will maintain its temperature or even allow it to warm, generating a positive surface sensible heat flux. Upward heat flux from the soil is also a significant contributor to surface heating once the surface net radiation has reached near zero (Duynkerke, 1999). The combination of these processes is what leads to the development of a well-mixed fog layer, but this process normally takes many hours. However, the model appears to be simulating this process almost instantaneously – mature, well-mixed fog is formed within an hour of fog onset.

The radiative effect of the fog appears to be a key feature in the process, therefore as a sensitivity test we turn off the radiative effect of any fog which forms in the model (i.e. by setting the absorption and scattering coefficients for condensed water to zero, denoted "No Rad Cld"), shown in Figures 1–3. The screen temperature and sensible heat flux now show a very good agreement with observations, although the surface temperature now gets far too cold and no appreciable fog layer actually forms, i.e. the visibility remains quite high. This demonstrates that the radiative feedback is very important in both the development of the
fog and maintaining the surface temperature. Although fog does form due to the temperature dropping and relative humidity reaching 100%, the lack of any enhanced radiative cooling from the fog itself prevents further development and thickening of the fog. There are also differences in the temperature structure between the model and observations. The layer of enhanced cooling near the surface is ≈20 m deep in reality, whereas it sits entirely below the lowest model level in the model. This means that the gradient is much sharper, explaining the colder surface temperature for given (and approximately correct) free
atmospheric temperatures and screen level temperature. It is unclear whether this is a consequence of the fog interaction, or the model vertical resolution as discussed in Vosper et al. (2013).

Whilst having radiatively inactive fog is clearly unrealistic, the differences in temperature and heat-flux evolution from the control simulation suggests that perhaps the fog is too radiatively active. Guedalia and Bergot (1994) have shown that errors in the development of radiation fog can be caused by errors in the dew deposition – depositing too little dew onto the
surface leaves too much condensed water in the atmosphere, which in turn has a too strong radiative effect. Figure 4a shows observations of dew deposition from the instrument described in Price and Clark (2014), in comparison to the model. During the main period analysed (20 UTC to 04 UTC) the control simulation is in reasonable agreement with the observations – it is at the lower end of the observational range, but certainly within the uncertainty of the instrument. Before 20 UTC, the control simulation under-estimates the deposition, and this is likely to be due to hygroscopic absorption, a process not accounted for
in the model's surface latent heat flux parametrization, as discussed in Price and Clark (2014). There is also a very shallow

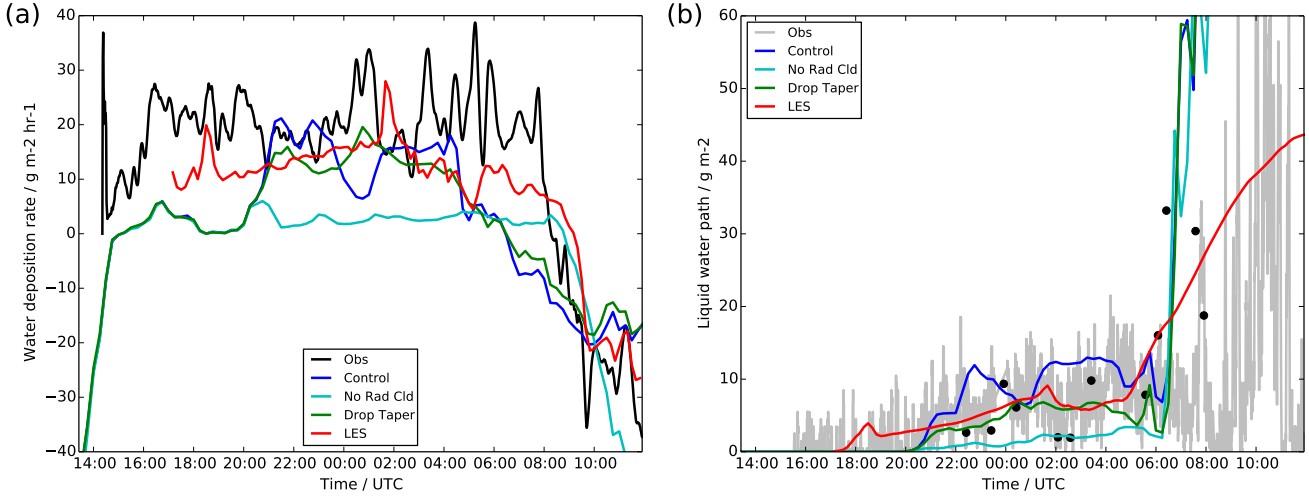

**Figure 4.** Time series of (a) surface water deposition rate, and (b) liquid water path, showing observations ((a) black; (b) tethered balloon: black dots, radiometer: grey line), LES (red) and UKV experiments: control (blue), radiatively inactive cloud (cyan) and modified droplet number (green).

and inhomogeneous fog present in reality at this time, which is not simulated by the model as it is only several meters deep and thus below the first model level, but will contribute to the observed deposition rate. The radiatively inactive fog by contrast shows almost no deposition throughout the simulation, and this is due to the fact that no appreciable fog layer forms in the simulation, and the dominant (model) process leading to surface dew deposition is sedimentation of condensed water onto the

5  surface.

Hence the dew deposition may be slightly under-estimated in the control simulation, but the effect of this will only become significant if it, in turn, leads to a significant over-estimate in the condensed water content. Figure 4b shows the liquid water path (LWP) and Figure 5a a profile through the fog layer from the tethered balloon. The profile shows the kind of variability which exists in the fog depth and water content, and is not indicative of any fog development between the two observation

10  times. The control simulation is in reasonable agreement with the observations for much of the night, although generally at the higher end of observed variability. The radiatively inactive fog is clearly too thin when compared to all available observations. Hence it appears that errors in dew deposition and the evolution of the condensed water content are not the main causes of error in the radiative effect of fog. However, the radiative effect of fog is not only controlled by the liquid water content, but also the number of cloud droplets, and therefore their size. Figure 5b shows that the number of droplets observed from the tethered

15  balloon is significantly less than those predicted by the model.

The model parametrization (Wilkinson et al., 2013) uses a fixed value of 75 cm$^{-3}$ for the droplet concentration at the lowest model level, tapering up to a value determined by the concentration of aerosol (Osborne et al., 2014) at 150 m. This tapering process is a pragmatic attempt at representing the fact that cloud drop numbers in fog tend to be lower than those observed higher up in the atmosphere (Gultepe et al., 2009; Price, 2011), however it appears that it is failing to represent some

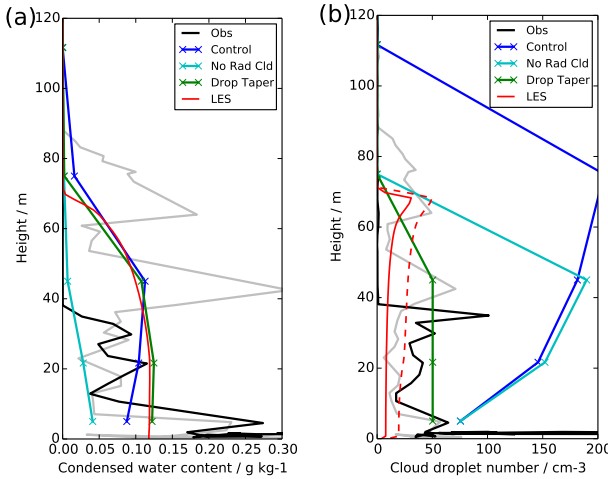

**Figure 5.** Profiles at 00.30 UTC showing observations (00 UTC: grey, 00.30 UTC: black), LES (red) and UKV experiments: control (blue), radiatively inactive cloud (cyan) and modified droplet number (green), showing (a) cloud liquid water content, and (b) cloud droplet number concentration. For the LES droplet number, we show only the activated droplets (solid) and all particles $> 2 \ \mu$m (dashed).

key features. The drop numbers in fog are believed to be low because, despite the abundance of aerosol from which to form cloud droplets near the surface, the lack of any appreciable updrafts in the near surface stable boundary layer results in weak supersaturations. This prevents the aerosol from being activated and therefore only relatively few aerosol droplets become hydrated. The observations also show the droplet numbers to be approximately constant throughout the fog layer, whilst the
model's (imposed) profile increases quickly with height from its surface value, due to the large value diagnosed higher up from the aerosol concentration.

    The abundance of small cloud drops in the model will increase the cloud absorptivity, making the fog optically thicker and more radiatively important. This will drive stronger radiative cooling from the fog top, enhancing the turbulence and causing the development of well-mixed fog. It will also increase the downwelling long-wave radiation at the surface, which will in turn
heat the surface and drive a positive sensible heat flux. However, observations of downwelling long-wave radiation (Fig. 6) are not available for much of the night, due to ice formation on the dome of the radiometer rendering the measurements unusable. The initial rise in downwelling long-wave radiation (between 20.30 and 23.00 UTC, before the ice formed) is representative of the fog, and it would not be unreasonable to assume that values remained near 270 $\mathrm{Wm}^{-2}$ throughout the night, perhaps increasing slightly as the fog developed. Therefore the model would appear to have excessive downwelling long-wave, but at
this point we shall appeal to the additional information that can be provided by the LES model to support this conclusion.

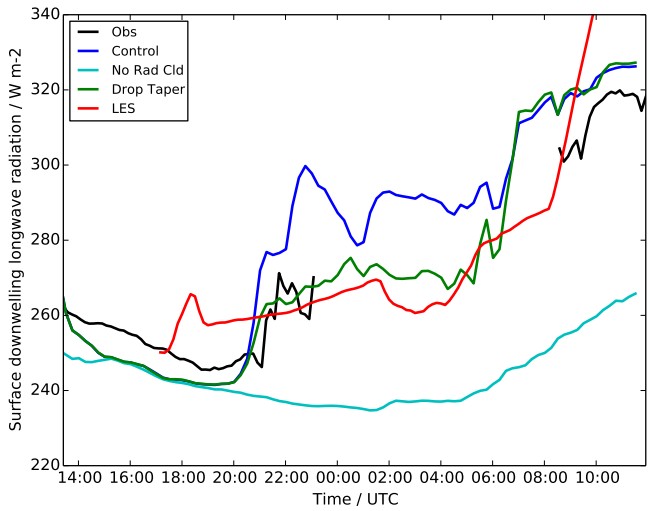

**Figure 6.** Time series of surface downwelling long-wave radiation, showing observations (black), LES (red) and UKV experiments: control UKV (blue), radiatively inactive cloud (cyan) and modified droplet number (green).

## 4 LES analysis

Figures 2-6 also showed results from the LES, which generally appear to be in good agreement with the observations, demonstrating that the LES is a representative proxy for reality. They also confirm that the low cloud droplet number (Fig. 5b) is responsible for a much lower downwelling long-wave radiation (Fig. 6) than simulated by the UKV. The LES also provides the opportunity to explore the mechanisms leading to cloud droplet activation (or lack thereof) in the fog, and therefore suggest improvements to the NWP model.

Figure 7 shows the near surface vertical velocity variance, which is very low in the observations and all models. Indeed it is safely below the 0.005 $m^2s^{-2}$ suggested in Price et al. (2018) for the threshold value above which fog will not form. What this implies is that the peak updraft speed driving aerosol activation is very low, typically <0.1 $ms^{-1}$. This is much lower than the typical updraft speeds found in 'normal' clouds (i.e. those which do not contain a rigid surface at their lower boundary). Because most models do not prognose or diagnose supersaturation, most aerosol activation parametrizations (e.g. Ghan et al., 1997; Morrison and Gettelman, 2008; West et al., 2014) link this updraft speed directly to the number of activated droplets. However, these parametrizations were not developed in the weak updraft regime of radiation fog, instead typically imposing a minimum updraft velocity or standard deviation of 0.1 $ms^{-1}$ or higher. Therefore, if any schemes like this are used to simulate aerosol activation in fog, they will systematically over-estimate the amount of aerosol activation, and therefore the cloud-droplet number, with inevitable consequences such as those discussed in Section 3.

For fog formation, updrafts are obviously not the only process that will activate aerosol and lead to the formation of cloud droplets. The fundamental process which is driving activation is the ambient supersaturation, driven by adiabatic cooling, which can be driven by updrafts but in fog is also driven by direct cooling of the atmosphere. The observed cooling rate during the

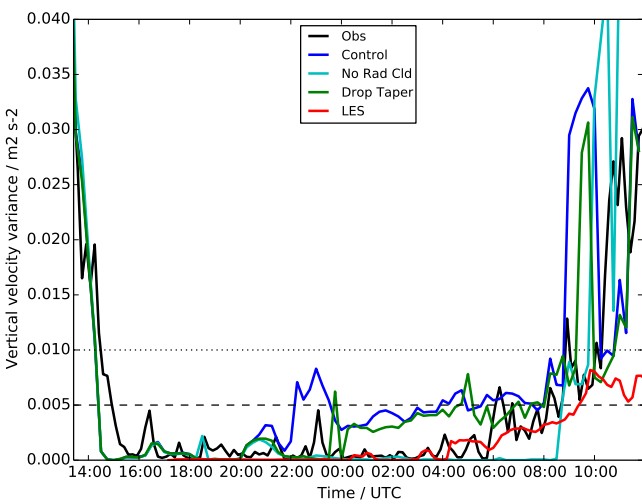

**Figure 7.** Time series of vertical velocity variance at screen level (2 m), showing observations (black), LES (red) and UKV experiments: control UKV (blue), radiatively inactive cloud (cyan) and modified droplet number (green). Also shown are the Price et al. (2018) value above which fog will not form (dashed) and minimum value used in most activation parametrizations (dotted).

first few hours of fog formation (Fig. 2a) is 1 Khour$^{-1}$, which would be equivalent to an updraft speed of 0.04 ms$^{-1}$ assuming a temperature lapse rate of 6.5 Kkm$^{-1}$. This is still significantly below the 0.1 ms$^{-1}$ minimum used for aerosol activation (even when added to the observed turbulent vertical velocities), demonstrating that typical parametrizations are not even indirectly representing the physical processes or amount of aerosol activation correctly. Further work is clearly warranted to develop
5 schemes which are appropriate for this regime.

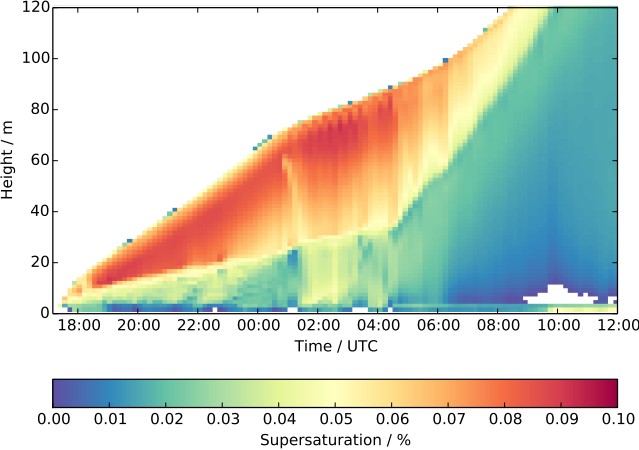

**Figure 8.** Time-height contour plot from the LES showing water-vapour supersaturation with respect to the saturation vapour pressure over liquid water.

Figure 8 shows the supersaturation as predicted by the LES throughout the simulation. The first point to note is that the values are very low, <0.1%, which is much lower than values typically found in clouds (0.3-1%, Ghan et al. 1997; 0.2-1.5%, Spracklen et al. 2011). Low supersaturation values are also supported by earlier observations in different fog campaigns. For example in the ParisFog campaign in France, the maximum effective supersaturation was found to be less than 0.05% with the average activation diameter of particles between 350 and 450 nm (Hammer et al., 2014). Similar findings were also obtained in the Po Valley, Italy, highlighting the role of aerosol chemical composition (Gilardoni et al., 2014).

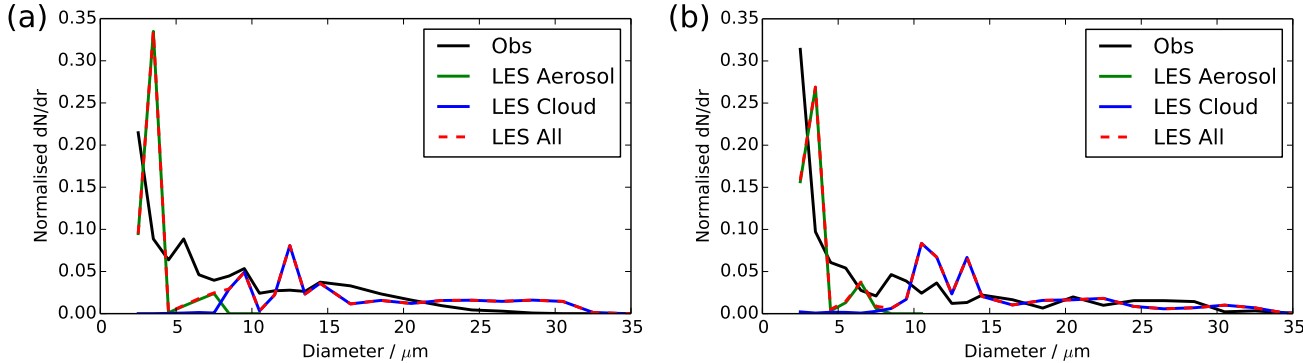

**Figure 9.** Cloud droplet size spectra at (a) 00.30 UTC, and (b) 03.30 UTC, showing observations (black) and LES (red). For the LES, we also show the number of activated drops (blue) and number of aerosol particles (green).

These low supersaturation values explain the very low levels of aerosol activation and therefore low cloud droplet number values. Figure 9 shows the size spectra of particles $> 2$ $\mu$m, which shows that there are large numbers of very small particles ($< 5$ $\mu$m) in both the observations and LES. It is likely that these particles are not activated cloud drops, but large aerosol particles which have absorbed water and swollen in size. We note here that we have excluded the lowest (1-2 $\mu$m) bin of the cloud-droplet probe from our analysis to avoid potential noise issues with large, dry aerosol affecting the counts. Because of the low supersaturation and therefore high activation diameter, hydrated aerosol particles within fog can grow larger than micrometer in size and can contribute up to 68% of total light-scattering during different fog periods (Hammer et al., 2014; Elias et al., 2015). However, most of these hydrated particles are still too small to considerably affect interaction with long-wave radiation after fog has developed. This explains how the visibility can become so low, and yet the fog remain optically thin and the boundary layer stable.

The LES allows us to analyse this further, because we can break the size spectra into their contributions from hydrated particles which are still in the LES aerosol classes, and particles which have been activated into the LES cloud-droplet classes, i.e. exceeded the critical size given by Köhler theory. As shown in Figure 9, almost all particles in the LES with diameter $< 6$ $\mu$m are wet aerosol, and only those with diameter $> 6$ $\mu$m are activated cloud drops. This distinction can also be seen in profiles of the cloud droplet number (Fig 5b, 10b), where we have shown both the LES results for all particles $> 2$ $\mu$m (as measured by the cloud droplet probe), and only those which are activated cloud drops. There is approximately a factor of

two difference between these two estimates of cloud droplet number, with the all-particle estimate comparing best with the observations. In the simulation, aerosol is assumed to be composed of ammonium sulphate, which might be more hygroscopic than the actual aerosol observed during IOP1. Less hygroscopic aerosol would have a lower growth factor within fog, and thus the transition from hydrated aerosol to activated fog droplets would be seen at a smaller diameter. However, less hygroscopic

aerosol would also activate less efficiently, thus increasing the critical dry diameter of activating particles. Thus we are confident that the small droplets seen in the observations are not fully activated, but significantly hydrated aerosol. This suggests that many observational estimates of fog droplet number, particularly in clean airmasses, could actually be over-estimating the number of activated or radiatively important (in the long-wave) droplets.

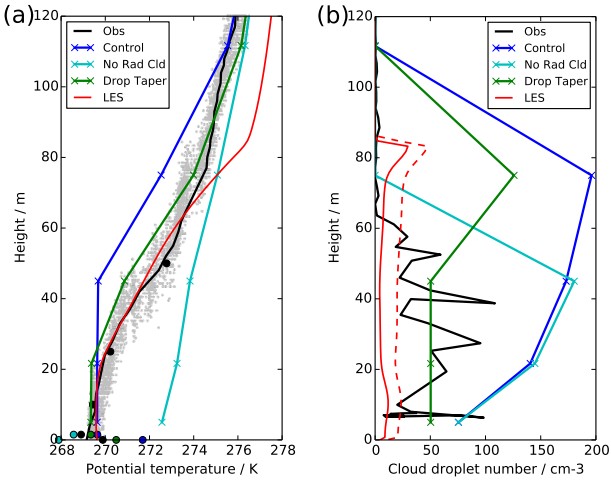

**Figure 10.** Profiles at 03.30 UTC showing observations, LES (red) and UKV experiments: control (blue), radiatively inactive cloud (cyan) and modified droplet number (green), showing (a) potential temperature, and (b) cloud droplet number concentration. Model profiles show model-level data (crosses) and diagnosed screen and surface level temperature (filled circles). For the LES droplet number, we show only the activated droplets (solid) and all particles $> 2\ \mu$m (dashed).

The supersaturation and turbulence levels remain remarkably constant throughout the night (up to 04 UTC), resulting in a

cloud droplet number concentration which also remains reasonably constant throughout the night (Fig 5b, 10b). This implies that the downwelling long-wave radiation (and optical depth) of the fog is mainly driven by its physical depth, which grows throughout the night (see Fig 8). The growth of the fog layer leads to a gradual, but continuous increase in the downwelling long-wave radiation throughout the night (Fig. 6), with the exception of a small decrease around 02 UTC. This occurs because of a sharp decrease in the background specific humidity (not shown) at around 80 m, and therefore the fog growth is temporarily

inhibited due to the entrainment of this drier air. This gradual rise in the downwelling long-wave radiation is responsible for the surface temperature ceasing to cool by 02 UTC and slowly warming throughout the rest of the night, which in turn drives a slow transition towards a well-mixed fog layer.

Figure 10 shows that by 03.30 UTC, the surface is warmer than the near-surface and the lowest 20 m of the atmosphere is well-mixed, with the top portion of the fog layer still stable. It is interesting to note that once the fog becomes adiabatic, the mixed layer temperature is very similar to that predicted by the control NWP model, although the NWP model mixed layer is much deeper because it has been developing for many hours. This can also be seen by the convergence of screen temperature in all models around 04 UTC in Fig. 2(a). After this time, the updraft speeds within the fog begin to grow, leading to increased droplet activation and a rise the observed and LES cloud droplet numbers. Fig 9(b) shows a small secondary peak in the size spectra around 10 $\mu$m developing due to the increasing turbulence, and the evolution of the spectra and cloud droplet numbers after this point is very similar to that already reported in Price (2011) and Tonttila et al. (2017), hence we do not focus on it here. The well-mixed layer continues to grow throughout the night, until the entire fog layer is well-mixed and turbulent by 08 UTC.

## 5 NWP model improvement

To test whether improving the representation of aerosol activation would improve the simulation of fog in the NWP model, we make a simple adjustment to the cloud droplet profile used in the UKV (denoted "Drop Taper"). Instead of tapering from a fixed value (75 cm$^{-3}$) at the surface to the aerosol dependant value at 150 m, we keep the droplet number fixed throughout the lowest 50 m of the atmosphere, before tapering towards the aerosol value above this. This is motivated by the observations and LES results, which show that the cloud droplet number is reasonably uniform throughout the fog layer. We also reduce the fixed value used in the lowest 50 m to 50 cm$^{-3}$, which is more representative of the observations, although still possibly too high. Clearly this choice is likely to be location and case specific, and require some adjustments for more polluted locations as discussed in Jayakumar et al. (2018). A universal parametrization would require some link to aerosol.

Figures 1-7 and 10 show the results from this simulation, and demonstrate that improvement is possible. The downwelling long-wave radiation (Fig. 6) now remains much lower throughout the night, which allows the surface to remain cool and the sensible heat flux remain near zero (Fig. 2). The model can maintain a stable atmospheric temperature profile for the early part of the night (Fig. 3a), and eventually it transitions to a well-mixed profile (Fig. 10a). Therefore, it would appear that by having a realistic estimate of cloud droplet numbers in fog we can significantly improve NWP fog simulations.

The mechanisms by which this improvement occur are twofold. Firstly, the reduced cloud droplet number increases the bulk sedimentation rate, which is calculated via Stokes Law (Wilkinson et al., 2013). This increases the rate at which condensed water is removed from the fog layer onto the surface, physically thinning the fog layer, i.e. the LWP is reduced. This is shown in Figure 4, where the LWP is approximately halved, yet the water deposition rate (which is almost entirely droplet settling) is largely unchanged. Secondly, the increased effective radius, which comes from reducing the cloud droplet number, reduces the cloud absorptivity to upwelling long-wave radiation (Slingo and Schrecker, 1982). This allows more radiation to be emitted to space, reducing the effectiveness of the fundamental process by which radiation fog develops – long-wave emission from the fog itself (Fig. 6), which is proportional to the absorption. Cooling from the fog top is less efficient at turbulence production, and downwelling long-wave radiation from the fog is less efficient at heating the surface (and thus enabling surface driven

turbulence to form), both of which inhibit the transition to well-mixed fog. The radiatively inactive cloud experiment is an extreme example of this effect.

Given the simple nature of the aerosol representation in the UKV, and current method of calculating cloud droplet numbers from this, the solution proposed above may actually be a suitable candidate for operational NWP implementation over the UK. To evaluate this, we run a month long trial of the full data assimilation and forecast system for February 2015 – 4 forecasts per day (00, 06, 12, 18 UTC), each 36 hours in length. This represented a typical UK winter, comprising periods of high-pressure with calm, potentially foggy conditions, and periods of westerly flow bringing low-pressure frontal systems across the country. The headline result is a small improvement across all measures of forecast skill (windspeed, temperature, cloud cover, precipitation, visibility). One of the more interesting changes was a non-negligible improvement in screen temperature forecast skill. Despite the fact that there were only relatively few foggy days during the month, the screen temperature error (similar to Fig. 2a) is so great when fog is present, that improvements to this can be seen in the mean temperature error across the entire month.

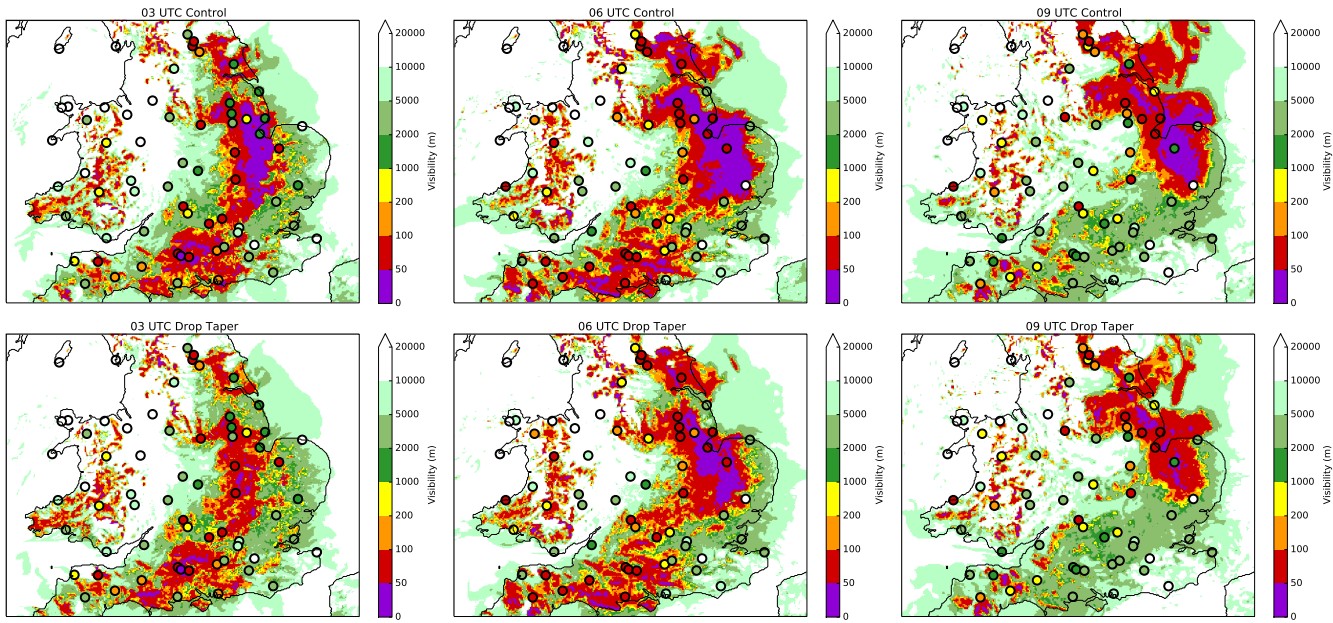

**Figure 11.** Visibility forecasts from 00 UTC on 10 February 2015 at three-hourly intervals from the UKV control (top) and with modified droplet number (bottom). Filled circles show observations.

The main improvements however are seen in surface visibility forecasts, such as those shown in Figure 11. The figure shows a widespread fog event over central and southern England on 10 February 2015. The fog onset occurs around 01 UTC in the observations, and both forecasts reproduce this reasonably well. However, once the fog is formed in the control simulation, it is instantly thick, well-mixed fog with visibilities <50 m, symptomatic of the problems discussed in Section 3. This is not observed in reality, with many stations reporting visibilities nearer to 100 m. The control simulation maintains these low

visibilities throughout the night as the fog moves across the country. By 09 UTC, the fog is dissipating in most places in reality, whilst the model retains a large area of thick fog with very low visibility.

With the revised droplet taper, the onset and dissipation phases of the fog event are improved. The fog onset is slower, with visibilities around 100 m forecast at 03 UTC, in general agreement with the observations. By 06 UTC, the fog has thickened, becoming well-mixed and with visibilities in places falling below 50 m. Whilst this is still too thick compared to the observations, it is certainly much better than the control. By 09 UTC, the fog has started to dissipate in the model, with visibilities rising, again in better agreement with the observations.

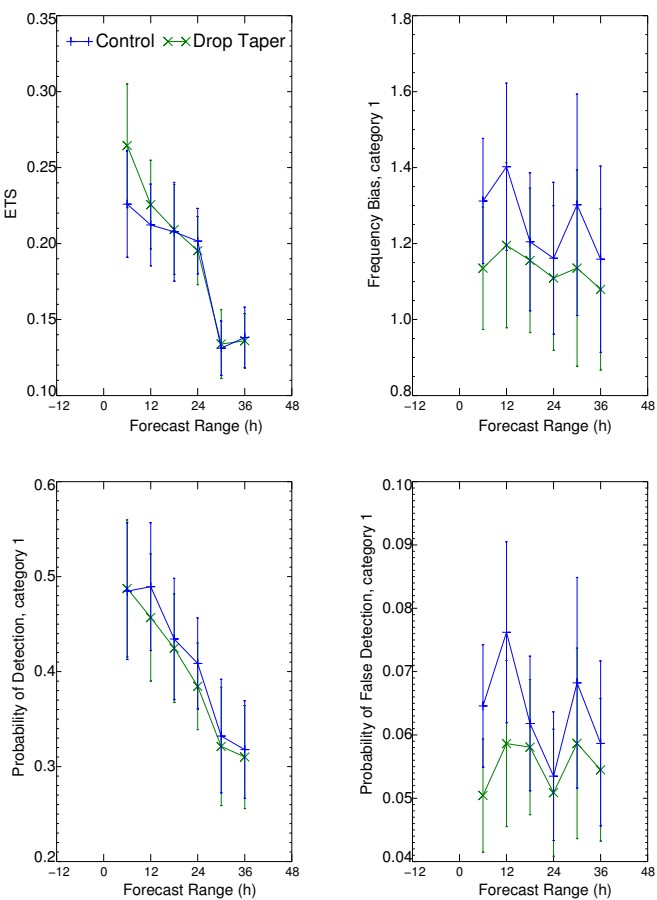

**Figure 12.** NWP model verification of visibility from 05/02/2015 to 05/03/2015 utilising all observations and forecasts (4 per-day) over UK land areas. Panels show the equitable threat score (ETS), frequency bias, probability of detection and probability of false detection, for a visibility threshold of 200 m, from the UKV control (blue) and with modified droplet number (green).

Figure 12 provides more quantitative support for these conclusions, presenting categorical verification of all forecasts in the trial period against all available surface observations. Following Mason (2003), we define a $2\times2$ contingency table such that $a$ is the number of hits, $b$ is the number of false alarms, $c$ is the number of misses and $d$ is the number of correct negatives.

We then determine the equitable threat score (ETS) as $(a - X)/(a - X + b + c)$ where $X = (a + b)(a + c)/(a + b + c + d)$, frequency bias as $(a + b)/(a + c)$, probability of detection (hit rate) as $a/(a + c)$ and probability of false detection (false alarm rate) as $b/(a + b)$. Perfect scores are 1 for the ETS, frequency bias and probability of detection, and 0 for the probability of false detection. In Figure 12 we consider the 200 m threshold for visibility, although results are similar at other thresholds (e.g. 1 km). As shown, the control simulation was over-forecasting low-visibility events, with a frequency bias $> 1$ and a probability of false detection $> 0$. Including the modified drop taper has clearly improved both of these metrics. Importantly, it has done this without significantly degrading the probability of detection, which remains largely unchanged, and therefore the ETS is improved. These results are consistent with the main results from IOP1, and the case study from this trial period presented in Fig. 11. The control simulation produces fog which is too thick, too fast, and tends to persist for too long, i.e. it over-forecasts. By improving the droplet numbers, this behaviour is improved, and can be seen in the statistical analysis.

Therefore, although not a panacea, the changes presented here certainly present a useful improvement in fog modelling skill and highlight that accurate representation of aerosol and its interaction with fog are key challenges for NWP skill. Future work should look to link the near surface droplet number to aerosol for a more complete and globally universal solution.

## 6 Conclusions

The manner in which atmospheric aerosol concentrations influence the number and size of fog droplets created in a typical radiation fog event, and how this influences the development of the fog, has been discussed. For given aerosol concentrations, the number of activated droplets is much lower in fog than usually found in clouds, due to the absence of strong updrafts to force cooling of air parcels. Instead, the activation of droplets relies on the very slow radiative cooling of the air, which creates small supersaturations, resulting in relatively few droplets becoming activated. The consequence of this for fog development is that fog can remain optically thin for many hours, relying on the physical growth of the fog layer to increase its optical depth and force the transition to well-mixed fog with a turbulent structure of its own. We can therefore summarise that key factors affecting the development of well-mixed fog include: (i) the amount of time available for development before sunrise, i.e. the length of the night and how soon after sunset the first fog forms, (ii) the speed with which the fog layer can deepen, strongly governed by the humidity profile – a moist environment will allow the fog to grow quicker and transition faster, and (iii) the amount of accumulation and coarse mode aerosol present for activation, as sensitivity tests with the LES demonstrated that considerably increasing the initial concentrations of larger aerosol lead to a faster transition to well-mixed fog. Non-local effects and advection are other factors that have not been discussed in this work but are likely to be important.

We have shown that accurate prediction of fog droplet number concentrations is crucially important to the accurate simulation of radiation fog. Excessive number concentrations, and therefore droplets which are too small, too radiatively important and do not sediment out fast enough, quickly lead to the fog becoming optically-thick and well-mixed, in stark contrast to the observations. This may appear to contradict the results of Maronga and Bosveld (2017), who find little effect on their simulations from changing the droplet number concentration. However, we believe their results are broadly consistent with our own. The simulations presented in Maronga and Bosveld (2017) transition rapidly to well-mixed fog, i.e. within 30 minutes of the first

onset of fog. Therefore, their simulation is very similar to our control simulation, and their sensitivity studies explore a range of droplet numbers (100-200 $cm^{-3}$) similar to our control simulation. Our observations and LES show drop numbers much lower than this range ($< 50$ $cm^{-3}$), and it is only when we reduce the drop numbers to this value that we achieve a simulation with a slow transition to well-mixed fog. It is also worth noting that the observations on which the Maronga and Bosveld (2017) sim-

ulations are based also present a slow transition to well-mixed fog (approximately 5 hours from the first onset), and so it would be interesting re-run their case with a much lower droplet concentration to see if the simulated transition can be improved.

This threshold-type behaviour occurs because once the fog has become optically thick, further changes to the cloud droplet number do little to affect the surface radiation balance and therefore fog development. It is only when the model can simulate fog in an optically-thin (i.e. upwelling and downwelling long-wave radiation not in balance) and turbulently stable state, that

the droplet number provides a much bigger control as it can determine when the transition between these states occurs. We believe that these results and suggested model improvements should be applicable to any NWP model, provided that the parametrizations of droplet settling and long-wave cloud absorption correctly depend on the droplet number/size, and so are keen to conduct an intercomparison of NWP models for a fog case like this to investigate these (and other) sensitivities.

Whilst we have shown that a simple approach to representing low droplet numbers in fog can lead to useful improvements

in NWP skill, this work has highlighted a key issue which has been overlooked in many NWP and climate models – how aerosol activation and droplet numbers in fog are calculated. More complex parametrizations of aerosol activation, such as Abdul-Razzak and Ghan (2000), are based around the strength of updrafts. Yet in this paper, we have shown that the minimum updraft speed often used in these parametrizations is considerably larger than those found in radiation fog, and the radiative cooling of the atmosphere is not sufficient to account for this discrepancy. Therefore it is likely that most models using these

types of activation parametrization produce excessive aerosol activation in their lowest model levels.

Poor representation of aerosol activation has notable consequences for climate models and their simulation of the aerosol effective radiative forcing of the climate system (Myhre et al., 2013). Figure 13(a) shows the in-cloud mean cloud droplet number concentration in the lowest model level of a 20 year present day climate simulation using the Met Office climate model HadGEM3-GA7 (Walters et al., 2017a). Throughout most of the northern hemisphere land masses, when fog is present, the

predicted droplet numbers are in excess of 150 $cm^{-3}$, with peak values over 250 $cm^{-3}$. This model is typical of many climate models in using the Abdul-Razzak and Ghan (2000) activation parametrization with a minimum updraft speed of 0.1 $ms^{-1}$. Based on the results of Section 4, Fig. 13(b) presents a sensitivity test where this minimum value is reduced to 0.01 $ms^{-1}$. As shown, there is a widespread reduction in the fog droplet number, in many regions by 50-100 $cm^{-3}$.

Figure 13 demonstrates that an artificial numerical minimum in the aerosol activation code is actually responsible for a large

proportion of near surface aerosol activation in HadGEM3-GA7. There is also potentially a significant impact on the climate system from this process, as the regions where the activation is happening are also predominantly the regions where the aerosol concentrations have grown most significantly throughout the 20th century. To quantify this, we re-run the simulations with pre-industrial aerosol concentrations, allowing us to estimate the effective radiative forcing following the method of Andrews (2014). The results show that reducing the minimum updraft speed reduces (i.e. makes less negative) the aerosol effective

radiative forcing by 0.1 $Wm^{-2}$. This is not an insignificant change, from a process which has been largely unconsidered by

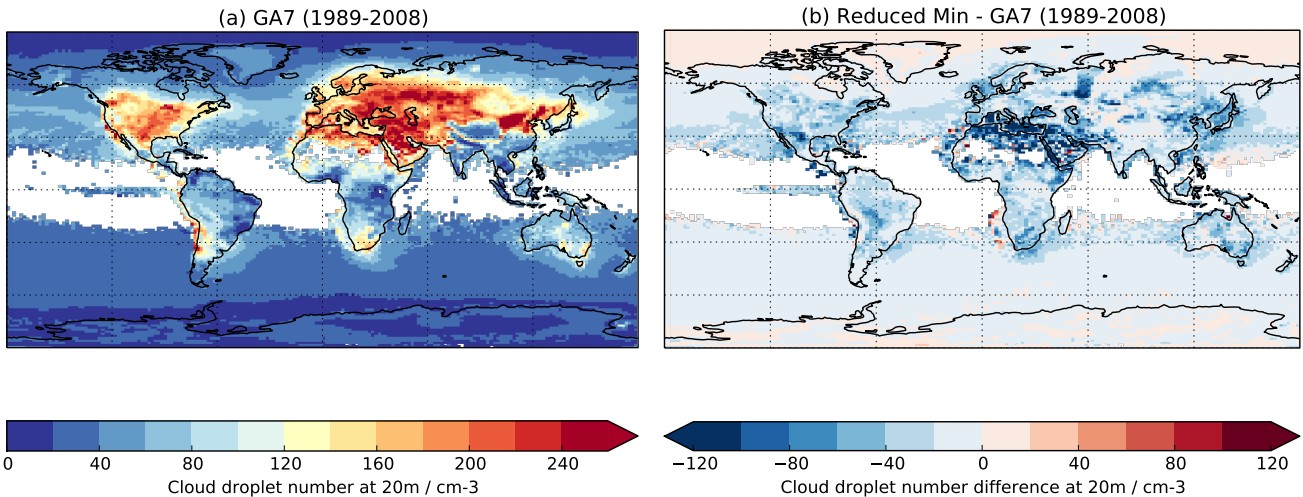

**Figure 13.** (a) 20 year mean cloud droplet number concentration at the lowest model level (20 m) from an atmosphere only simulation of HadGEM3-GA7. (b) The difference from this when reducing the minimum updraft speed used for aerosol activation from 0.1 ms$^{-1}$ to 0.01 ms$^{-1}$.

climate models to date. Whilst it is beyond the scope of the current paper to evaluate these changes to HadGEM3-GA7 (this will be done in Mulcahy et al., 2018), it is clear that the representation of fog processes in climate models is an area warranting further attention with high priority. Vautard et al. (2009) have shown that recent reductions in fog occurrence over Europe are a significant contributor to surface temperature warming, and therefore missing or mis-representing this process in climate

5 projections is problematic.

**Data availability**

All data used in this study is available from the authors upon request.

*Competing interests.* The authors declare that they have no conflict of interest.

*Acknowledgements.* IB and JP thank the UK Civil Aviation Authority for providing some funding towards this research. IK, HK and SR are

10 supported by the Academy of Finland (project number 283031, and the Centre of Excellence in Atmospheric Science, no. 272041). We also thank Adrian Hill for many inspiring discussions on aerosol activation in fog.

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
