# Peer review of "Aerosol-fog interaction and the transition to well-mixed radiation fog"

_Atmospheric Chemistry and Physics, 2017_

## Referee Comment (RC1) · Anonymous Referee #1 · 24 Oct 2017

**Review of the paper entitled**
**"Aerosol-fog interaction and the transition to well-mixed radiation fog"**

by Ian Boutle, Jeremy Price, Innocent Kudzotsa, Harri Kokkola and Sami Romakkaniemi

**manuscript number ACP-2017-765**

[Figure]

**[ACPD]**
*Decision* : This paper need more work before it goes to publication. Accepted with MAJOR improvements.

This paper addresses the difficult topic to evaluate the influence of aerosol for the numerical simulation of radiation fog. The results are based on one case observed at Cardington-UK during IOP1 of LANFEX field experiment and on one month of assimilation - forecast for February 2015. The subject of the manuscript is interesting as radiation fogs are not well known, but the actual scope of the manuscript is not so well defined. I have concerns about the methods used and I wonder about the value added by this study with respect to the existing bibliography.

Therefore I can not recommend publication of this paper without extensive and major revisions.

I would like to see the paper again once major modifications are done.

**major remarks** :

1. **Demonstrate the usefulness of using 3 types of models (LES - NWP - climate model)** :
   Previous studies on microphysical processes were done with NWP or 1D models or were the results of field experiments measurements. Please justify the use of LES to demonstrate the impact of aerosol on fog. What are the added values of LES study with respect to 1D model? This point is essential for the publication of this work.
   The impact of cloud droplet number with NWP should be evaluated in a statistical way : How has this modification improved the fog forecast? This evaluation could
be done with LANFEX data which provide many cases of "stably stratified fog" cases. Your conclusions are too speculative and need to be demonstrated.

2. **Validate the microphysical parameterization** :
   Your work is based on the use of a specific microphysical parametrization. In my opinion, the main characteristics of this parametrization should be detailed in the revised manuscript. Particularly, the dependence of activation parametrization with respect to radiative cooling and turbulent processes should be explained. This microphysical parametrization should also be validated for fog cases, with comparison with observations from LANFEX for example.

3. **Validate the numerical model used and particularly the frost-dew deposition** :
   The dew and frost deposition play a key role during the formation phase of fog and particularly in the transition to well-mixed radiation fog (eg Guedalia and Bergot 1994). Frost deposition could prevent the formation of dense fog despite radiative cooling. It is necessary to demonstrate that this process is correctly simulated. Otherwise, your modification may simply compensate the errors in the estimate of deposition by the model. It is also absolutely necessary to validate the model used for turbulent processes, radiation and soil-atmosphere exchanges.

4. **Contribution of this study with respect to bibliography** :
   Another shortcomings of the current manuscript is that it does not cite available literature. For example papers from Bott (1991) or Rangognio et al. (2009) have studied in detail the effect of aerosol on radiation fog. It needs to be shown what new results does this work provide compared to those already published. What are the differences in the model, observation and methods used? What are the differences in results found? You say in conclusions that "key factors affecting the development of well-mixed fog include :"
   "(i) the amount of time available for development before sunrise" : the point is
well-known and has also be demonstrated by numerous studies. Please cite the current literature and please evaluate your contribution with respect to existing work.

"(ii) the speed with which the fog layer can deepen , strongly governed by humidity profile" : please evaluate your contribution for this well-known result.

"(iii) the amount of accumulation and coarse mode aerosol for activation" : this point is in contradiction with the LES study of Maronga and Bosveld (2017) on a Cabauw case which say in abstract that "the choice of droplet number concentration ... has a high impact on the liquid water content within the fog layer but a rather small effect on its life cycle". Please elaborate.

---

## Referee Comment (RC2) · Anonymous Referee #2 · 26 Oct 2017

Review of the manuscript acp-2017-765 Aerosol-fog interaction and the transition to well-mixed radiation fog by Ian Boutle et al.

Summary In this paper the authors study the impact of the interaction of aerosol and fog formation from both a modelling as well as observational perspective. They select a case study from the LANFEX campaign that took place at Cardington (UK). That case is modelled with a large eddy simulation model in which the aerosol concentration is varied. It is seen that default (which are excessive in this case) aerosol concentrations result in the emergence of optically thick fog too early. It is shown that a realistic lower concentration results in a better representation of life cycle of the fog, and this modification is subsequently implemented in an operational model as well as a climate model. As fog is a critical weather phenomenon that is challenging to forecast, this

study is a welcome contribution on the understanding of the fog.

Remarks: Abstract: -"Improvements to the representation of cloud droplet concentration" -> it is more correct to say that you implemented a reduction of the concentration.

Structure: In general the paper is well written, though I recommend the paper structure should be a bit more clearly presented in the beginning of the paper. The paper contains a discussion of observations, LES, and NWP model and a climate model. Some of these appearances appear a bit as a surprise or are only very briefly introduced, which give the reader a bit the uncertain feeling like "where do we go?". Hence the authors should better introduce why the chain of models presented in necessary to answer the research questions. Just some more details would be appreciated, which would also help to ensure reproducibility.

Figures: I suggest to plot observations in dots and model results as lines so they are more easy to distinguish without reading the caption twice.

Synthesis: I encourage the authors to strengthen the discussion section. Somehow I have the feeling the paper has a bit the nature of a technical report that present the impact of changes in model settings, that are in itself of course valuable, but the synthesis how the current results relate to other studies could be strengthened. At least I am aware of other studies that report on a much smaller sensitivity on the microphysical settings than presented here. As such a more complete synthesis would be valuable.
* * *

---

## Author Comment (AC1) · 14 Nov 2017

**Aerosol-fog interaction and the transition to well-mixed radiation fog:**
**Response to reviewer 1**

Ian Boutle, Jeremy Price, Innocent Kudzotsa, Harri Kokkola, Sami Romakkaniemi

November 14, 2017

We thank the reviewer for their thorough review, and will update the manuscript accordingly. We have provided detailed responses to all points below, and would welcome any further comments the reviewer may have whilst the discussion phase is still open.

**1. Demonstrate the usefulness of using 3 types of models (LES - NWP - climate model) : Previous studies on microphysical processes were done with NWP or 1D models or were the results of field experiments measurements. Please justify the use of LES to demonstrate the impact of aerosol on fog. What are the added values of LES study with respect to 1D model? This point is essential for the publication of this work.**

The ultimate purpose of this work is to improve NWP or climate simulation of fog events. To do this, we have chosen two sources of information to aid our understanding - observations and LES. The main reason we have done this is because neither source alone can provide all of the information we need. The observations are obviously closest to the truth, but there are gaps in the observational dataset, both in terms of quantities measured, and spatial/temporal representivity of the data. Therefore, to assist with the interpretation and understanding of the observations, we have supplemented them with LES data.

The main information we require from the LES to support our conclusions is the downwelling longwave radiation from the period where the radiometer was frozen (Fig. 6). The partitioning between hydrated aerosol and activated cloud droplets comprising the size spectra in Fig. 9 is an additional piece of interesting information provided by the LES, for which we do not have observations. Finally, the mean profiles from the LES are closer to what we would expect an NWP or climate model to simulate. The observations are point samples and can be subject to considerable variability (e.g. the difference in consecutive tethered balloon profiles shown in Fig. 5). The ability to explore this variability in the LES allows us to determine whether changes seen in the observations are due to evolution of the fog layer or natural

variability.

We feel that LES (as opposed to a 1D model) is the best tool for this job, because it is least reliant on parametrizations. The key focus of this paper is the coupling between aerosol processes and turbulence. The LES is able to resolve the important scales of turbulence, simulating the variability that exists within a fog layer forming over a spatially homogeneous terrain. We can therefore couple the microphysics directly to the turbulent flow, allowing for the activation in updrafts and supersaturated regions, without having to rely on sub-grid estimates of turbulent quantities from a parametrization. A 1D model would be providing these from a turbulence parametrization, which would itself, most likely have been derived from LES simulations, and therefore we are avoiding this intermediate step.

We will include this discussion and motivation in the revised paper.

**The impact of cloud droplet number with NWP should be evaluated in a statistical way : How has this modification improved the fog forecast? This evaluation could be done with LANFEX data which provide many cases of "stably stratified fog" cases. Your conclusions are too speculative and need to be demonstrated.**

We had not initially wanted to present a detailed analysis of the NWP model verification, as the key focus of the paper is a process modelling study which should be applicable to any NWP system. We merely wanted to illustrate with Figures 11 and 12 that lessons learned from the process modelling study do provide useful benefit in an NWP modelling system. We have included in Figure 1 and 2 the statistical results of the month-long trial of the full data assimilation and forecast system. These plots incorporate all data from UK land stations to verify the forecasts (4 per-day at 00, 06, 12 and 18 UTC) for their 36 hour forecast range. There are therefore many more forecasts and observations in this data-set than in, for example, the LANFEX archive.

Figure 1 shows categorical statistics for observed and forecast visibilities of 200 m and 1 km. Following Mason (2003), we define a 2×2 contingency table such that $a$ is the number of hits, $b$ is the number of false alarms, $c$ is the number of misses and $d$ is the number of correct negatives. We then determine the Equitable Threat Score (ETS) as $(a - X)/(a - X + b + c)$ where $X = (a + b)(a + c)/(a + b + c + d)$, Frequency Bias as $(a + b)/(a + c)$, Probability of Detection as $a/(a + c)$ and Probability of False Detection as $b/(b + d)$. Perfect scores are 1 for the ETS, Frequency Bias and Probability of Detection, and 0 for the Probability of False Detection. As shown, the "Control" model was over-forecasting low-visibility events, with a Frequency Bias > 1 and a Probability of False Detection > 0. Including the modified drop taper has clearly improved both of these metrics. Importantly, it has done this without significantly degrading the Probability of Detection, which remains largely unchanged, and therefore the ETS is improved. These results are consistent with the main results from LANFEX IOP1, and the case study from this trial period presented in Fig. 11. The model produces fog which is

[Figure]

Figure 1: NWP model verification of visibility from 05/02/2015 to 05/03/2015 utilising all observations and forecasts (4 per-day) over UK land areas. Panels show the Equitable Threat Score (ETS), Frequency bias, Probability of Detection and Probability of False Detection for visibility thresholds of 200 m (left) and 1 km (right), for experiments "Control" and "Drop Taper".

too thick, too fast, and tends to persist for too long, i.e. it over-forecasts. By improving the droplet numbers, this behaviour is improved, and can be seen in the statistical analysis.

Fig 2 shows the mean error and root-mean-squared (RMS) error for visibility and screen-level temperature forecasts. The results show improved RMS errors when including the modified droplet taper, and a slightly degraded mean error, which is again consistent with the other results presented. The effect of the revised drop taper is to reduce low visibility events, which will lead to an increase in mean visibility. When the model already has a high bias in mean visibility, this will inevitably get worse. However, the high bias in mean visibility comes mainly from high visibility events (visibilities > 10 km), and therefore by over-forecasting low visibility events we were compensating for this error in a mean sense. We have now removed one half of this compensating error.

The screen temperature shows improvements to both the mean and RMS errors. The improvement to the mean error tends to be removing a cold bias in the model, showing that the most significant effects on temperature verification are from allowing the fog to dissipate earlier during the morning period and allowing the model to warm up faster during the day.

[Figure]

Figure 2: NWP model verification of visibility (left) and screen-level temperature (right) from 05/02/2015 to 05/03/2015 utilising all observations and forecasts (4 per-day) over UK land areas. Panels show Mean Error and Root-Mean-Squared Error for experiments "Control" and "Drop Taper", and the difference of Drop Taper from Control.

In the revised manuscript, we will replace Figure 12 with some of this statistical evidence, to add further support to our results. The manuscript will then contain a detailed process modelling study, an independent example case, and statistical evidence from many cases, all showing the same behaviour and model improvements.

**2. Validate the microphysical parameterization : Your work is based on the use of a specific microphysical parametrization. In my opinion, the main characteristics of this parametrization should be detailed in the revised manuscript. Particularly, the dependence of activation parametrization with respect to radiative cooling and turbulent processes should be explained. This microphysical parametrization should also be validated for fog cases, with comparison with observations from LANFEX for example.**

We assume here you are referring to the microphysical parametrization in the LES model. In UCLALES-SALSA the aerosol size distribution is represented with a sectional method using 10 size classes (bins), and water condensation on particles in different bins is calculated by numerically solving the condensation equation at every time step in every grid point. Thus we

are able to explicitly simulate how the radiative cooling (which increases water saturation ratio S) and turbulence (updrafts enhance S and downdrafts decrease S) affect the water saturation ratio and how this affects the size of aerosol particles. As soon as the first particles grow above their critical size given by Kohler theory, they will be counted as cloud/fog droplets and technically transferred into separate cloud droplet size distribution. Water condensation on droplets is also solved through the condensation equation, and condensation on both aerosol particles and droplets is affecting the water amount in the gas phase. Thus we do not employ any such parameterization for droplet activation which are traditionally used in large eddy or other atmospheric modelling applications, but instead we simulate the actual supersaturation and growth of aerosol particles into droplets. We have tested with a more detailed cloud parcel model (e.g. Kokkola *et al.*, 2008) that the growth of particles into cloud droplets is solved with a good accuracy in SALSA for the case of adiabatic air parcels with different updraft velocities.

In LANFEX we are missing the aerosol information to validate the microphysical scheme, but supersaturation simulated by UCLALES-SALSA agrees well with measured aerosol activation in the existing field campaigns already cited in the manuscript. We have already provided qualitative comparison of model results against similar observations to LANFEX in Tonttila *et al.* (2017), and detailed analysis of model simulations of radiation fog in Maalick *et al.* (2016). The model was found to capture the main characteristics of radiation fog formation, and to provide similar results to other LES models. Finally, it's worth noting that the performance of the LES against observations for this case (as presented in Figs. 2-10) is very good, and has been achieved without any specific tuning of the LES.

We will make this more clear in the manuscript and add references into the recent literature where the lack of radiative cooling is found problematic for activation parametrizations, and explain how this is avoided in our case.

**3. Validate the numerical model used and particularly the frost-dew deposition : The dew and frost deposition play a key role during the formation phase of fog and particularly in the transition to well-mixed radiation fog (eg Guedalia and Bergot 1994). Frost deposition could prevent the formation of dense fog despite radiative cooling. It is necessary to demonstrate that this process is correctly simulated. Otherwise, your modification may simply compensate the errors in the estimate of deposition by the model.**

We again assume here you are referring to the LES model. Dew deposition is clearly an important process, and something we plan to study more as part of the LANFEX project, because this is likely to be an area in which all models are deficient. During LANFEX, we have measurements of dew deposition available from the instruments described in Price and Clark (2014), in addition to typical near surface humidity measurements.

Figure 3 shows observations of dew deposition from IOP1, and equivalent simulated quan-

[Figure]

Figure 3: Time series of surface water deposition rate (i.e. dew, left) and screen level (1.5 m) specific humidity (right) from observations (black), LES (red) and UKV experiments: control (blue), radiatively inactive cloud (cyan) and modified droplet number (green)

tities from all the models analysed. During the main period analysed (20 UTC to 04 UTC) the LES (and NWP models) are in reasonable agreement with the observations – they are at the lower end of what the observations are showing, but certainly within the uncertainty of the instrument. During the early stages (before 20 UTC) the models appear to be under-estimating the deposition, and this is likely to be due to hygroscopic absorption, a process not accounted for in the model's surface latent heat flux parametrization, as discussed in Price and Clark (2014). During the latter stages (after 04 UTC), the models again appear to be under-estimating the deposition, and this is due to a compensation between the fog droplet sedimentation, which is still in good agreement with the observations, and the latent heat flux, which is now positive causing the surface to lose water.

As discussed in Guedalia and Bergot (1994), the key effect dew deposition has on the fog lifecycle comes through its modification of the near-surface specific humidity, which is also shown in Figure 3 and profiles at various times are shown in Figure 4. The observations show a reduction in specific humidity from 5 gkg$^{-1}$ to 3 gkg$^{-1}$ during the development stages of the fog event, and this is reasonably well simulated by the models. The profiles presented in Fig. 4 show how this reduction in water vapour grows through the lower levels of the atmosphere, up to 80 m by 03.30 UTC. Again, this reduction is well simulated by all of the models, except for "No Rad Cld" which it not a particularly realistic simulation. This reduction in specific humidity is consistent with water transfer into the condensed state (the fog), for which observations show that the water contents are well simulated by the models (Figs 4 and 5a in the paper), and deposition onto the surface (Fig. 3).

Therefore, we are confident that this process is being correctly simulated in the model, and not leading to erroneous fog development. However, this is also clearly an uncertain process requiring further work. Indeed it will interact with the aerosol effects discussed in this paper.

[Figure]

Figure 4: Profiles of specific humidity at 17.30 UTC (left), 22.30 UTC (middle) and 03.30 UTC (right), showing observations (mast: black circles, radiosonde: black line), LES (red) and UKV experiments: control (blue), radiatively inactive cloud (cyan) and modified droplet number (green). Model profiles show model-level data (crosses) and diagnosed screen level data (filled circles).

The droplet sedimentation flux is inversely proportional to the number of fog droplets, and therefore deposition will be high in situations with low fog droplet number, such as IOP1. In situations with higher droplet number, the deposition will be much lower. This is another mechanism (in addition to the radiative mechanism discussed in the paper) which will lead to strong sensitivity of fog processes to aerosol.

We shall add some of this discussion to the revised manuscript, to demonstrate that the dew deposition is realistic in the models, but is another area of uncertainty which requires further study.

**It is also absolutely necessary to validate the model used for turbulent processes, radiation and soil-atmosphere exchanges.**

As mentioned in response to point 2, Maalick *et al.* (2016) and Tonttila *et al.* (2017) have analysed and validated the LES for radiation fog cases similar to those presented here. The LES model used also has an extensive pedigree for modelling turbulence and the interaction of radiation with other types of clouds (e.g. Stevens *et al.*, 1999, 2005), where it has been shown to be of similar quality to any other commonly used LES code. Finally, the comparison of LES results to turbulence and radiation observations for the case presented (e.g. Figs. 2, 3, 7) is very good. As the surface temperature is specified throughout the simulation, there are no soil-atmosphere exchanges or land surface feedbacks within the LES. This allows us to avoid the large uncertainty in simulation of land surface processes, and choice of initial conditions for soil heat and moisture content, something which was pointed out by Maronga and Bosveld (2017) as one of the main uncertainties in the formation time of fog.

We shall clarify these points in the revised manuscript.

**4. Contribution of this study with respect to bibliography : Another shortcomings of the current manuscript is that it does not cite available literature. For example papers from Bott (1991) or Rangognio et al. (2009) have studied in detail the effect of aerosol on radiation fog. It needs to be shown what new results does this work provide compared to those already published. What are the differences in the model, observation and methods used? What are the differences in results found?**

We agree that not citing the Bott (1991) paper was an over-site on our part, and this shall be corrected. The Bott (1991) study is indeed very complimentary to our own, demonstrating the effect of aerosol concentrations on the development of fog in a 1D model. Our work has reproduced and analysed in detail the mechanisms for this effect in two independent models (LES and NWP), utilising different parametrizations of microphysics, radiation and turbulence. Furthermore, we have provided observational evidence for this effect in a real case study, and perhaps most importantly have provided a simple method of incorporating this effect in an operational NWP system, demonstrating the benefits of including it on forecast quality. This latter point is key – it is 26 years since the study of Bott (1991) and yet very few operational NWP models include the interaction of aerosols with fog in their systems, despite the obvious importance which is noted by Bott (1991) – "in the morning, the urban fog is not dissipated by the solar radiation". This discussion shall be included in the revised manuscript.

Whilst it is an interesting paper, we are reluctant to cite the study of Rangognio *et al.* (2009) because it was never published, due to errors discovered in the numerical model. It is therefore difficult to fully assess any similarities or differences with our work.

**You say in conclusions that "key factors affecting the development of well-mixed fog include :" "(i) the amount of time available for development before sunrise" : the point is well-known and has also be demonstrated by numerous studies. Please cite the current literature and please evaluate your contribution with respect to existing work. "(ii) the speed with which the fog layer can deepen , strongly governed by humidity profile" : please evaluate your contribution for this well-known result.**

We don't believe we have claimed to make any contribution to these well known results, we were merely trying to synthesise for the reader all the key processes governing the development of well-mixed fog. We shall clarify this in the revised text.

**"(iii) the amount of accumulation and coarse mode aerosol for activation" : this point is in contradiction with the LES study of Maronga and Bosveld (2017) on**

**a Cabauw case which say in abstract that "the choice of droplet number concentration ... has a high impact on the liquid water content within the fog layer but a rather small effect on its life cycle". Please elaborate.**

The comparison with Maronga and Bosveld (2017) is interesting, and certainly something we should discuss in the manuscript. The case presented by Maronga and Bosveld (2017) transitions to well-mixed fog almost instantly in the model - this can be seen in their Fig. 2 where the visibility reduces to 100 m at all vertical levels (2, 10 and 20 m) within 30 minutes, and Fig. 3 where the downwelling longwave radiation rapidly increases to match the upwelling longwave radiation at the time of modelled fog onset. Therefore, their simulation is very similar to our "Control" run. Their sensitivity studies to fog droplet number encompass the range 100-200 cm$^{-3}$, which is approximately the vertical variability in droplet number shown by our control run (Fig. 5). Our observations and LES show drop numbers much lower than this range ($< 50$ cm$^{-3}$), and it is only when we reduce the drop numbers to this value that we achieve a simulation with a slow transition to well-mixed fog. It's also worth noting that the case investigated by Maronga and Bosveld (2017) probably also has a slow transition to well-mixed fog in reality, with the visibility falling to 100 m, at 2 m height, 2 hours before it does at 10 m, and 3 hours before it does at 20 m, in the observations. This is in stark contrast to the model which achieves this vertical development in 30 minutes. It would be interesting to re-run the simulations of Maronga and Bosveld (2017) with a lower droplet concentration, to see if this transition can be improved.

Another point, which we should make clearer, is that the feedback between aerosol and fog development is dependant on longwave scattering in the radiation parametrization, a process which is often ignored in NWP models (Edwards and Slingo, 1996). And indeed if we turn off this process in our NWP model, we do not see such strong sensitivity of the fog development to the aerosol concentration. It may be that non-inclusion of longwave scattering is a reason for reduced sensitivity seen in other studies.

We will add discussion and clarification of these points to the revised manuscript.

**References**

Bott, A. (1991). On the influence of the physico-chemical properties of aerosols on the life cycle of radiation fogs. *Boundary-Layer Meteorol.*, **56**, 1–31.

Edwards, J. M. and Slingo, A. (1996). Studies with a flexible new radiation code. I: Choosing a configuration for a large-scale model. *Q. J. R. Meteorol. Soc.*, **122**, 689–719.

Guedalia, D. and Bergot, T. (1994). Numerical Forecasting of Radiation Fog. Part II: A Comparison of Model Simulation with Several Observed Fog Events. *Mon. Weather Rev.*, **122**, 1231–1246.

Kokkola, H., Korhonen, H., Lehtinen, K. E. J., Makkonen, R., Asmi, A., Järvenoja, S., Anttila, T., Partanen, A.-I., Kulmala, M., Järvinen, H., Laaksonen, A., and Kerminen, V.-M. (2008). SALSA – a sectional aerosol module for large scale applications. *Atmos. Chem. Phys.*, **8**, 2469–2483.

Maalick, Z., Kühn, T., Korhonen, H., Kokkola, H., Laaksonen, A., and Romakkaniemi, S. (2016). Effect of aerosol concentration and absorbing aerosol on the radiation fog life cycle. *Atmos. Eviron.*, **133**, 26–33.

Maronga, B. and Bosveld, F. (2017). Key parameters for the life cycle of nocturnal radiation fog: a comprehensive large-eddy simulation study. *Q. J. R. Meteorol. Soc.*, **143**, 2463–2480.

Mason, I. B. (2003). Binary events. In I. T. Jolliffe and D. B. Stephenson, editors, *Forecast verification – a practitioner's guide in atmospheric science*, pages 37–76. Wiley.

Price, J. D. and Clark, R. (2014). On the measurement of dewfall and fog-droplet deposition. *Boundary-Layer Meteorol.*, **152**, 367–393.

Rangognio, J., Tulet, P., Bergot, T., Gomes, L., Thouron, O., and Leriche, M. (2009). Influence of aerosols on the formation and development of radiation fog. *Atmos. Chem. Phys. Discuss.*, **9**, 17963–18019.

Stevens, B., Moeng, C.-H., and Sullivan, P. P. (1999). Large-eddy simulations of radiatively driven convection: Sensitivities to the representation of small scales. *J. Atmos. Sci.*, **56**, 3963–3984.

Stevens, B., Moeng, C.-H., Ackerman, A. S., Bretherton, C. S., Chlond, A., de Roode, S., Edwards, J., Golaz, J.-C., Jiang, H., Khairoutdinov, M., Kirkpatrick, M. P., Lewellen, D. C., Lock, A., Müller, F., Stevens, D. E., Whelan, E., and Zhu, P. (2005). Evaluation of large-eddy simulations via observations of nocturnal marine stratocumulus. *Mon. Weather Rev.*, **133**, 1443–1462.

Tonttila, J., Maalick, Z., Raatikainen, T., Kokkola, H., Kühn, T., and Romakkaniemi, S. (2017). Uclales–salsa v1.0: a large-eddy model with interactive sectional microphysics for aerosol, clouds and precipitation. *Geosci. Model Dev.*, **10**, 169–188.

---

## Referee Comment (RC3) · Anonymous Referee #1 · 21 Nov 2017

**"Aerosol-fog interaction and the transition to well-mixed radiation fog"**

by Ian Boutle, Jeremy Price, Innocent Kudzotsa, Harri Kokkola and Sami Romakkaniemi

**manuscript number ACP-2017-765**

[Figure]

I would like to thank the authors for this very detailed response to my previous comments. You have did a very huge work and the impact of your research could be maximized. I hope that my comments would help you to improve your article.

1. **Demonstrate the usefulness of using 3 types of models (LES - NWP - climate model)** :
   I totally agree with you about complementarity between observations and LES. LES are a great tool to better understand the processes occuring during the fog life cycle. I also agree that the ultimate goal is to improve NWP simulations, and a statistical study could demonstrate that this goal is achieved. In my opinion, the statistical validation of NWP is very interesting and needs to be included in the revised version. However, for fog (rare event) I am not sure that probability of false detection ($b/(b+d)$) would be the best indicator of false alarm because $d >> b$. I prefer the false alarm rate : $b/((a+b)$

   I am not convinced by the usefulness of climate simulations. In my opinion, this part of the article makes the manuscript more confusing without added scientific values.

   For the LES, it would be useful to study in detail the variability found in LES simulations, and to validate it with observations (if available). Moreover, what is the impact of microphysics in this variability? I also have questions about activation processes in LES model. Given the time step used in LES study, I am not sure that a direct coupling between microphysics and LES updrafts (turbulent updrafts) is the best way to modelize the activation process. What is the representative time for activation processes? Is it compatible with the time step used in LES or with the lifetime of turbulent updrafts?

2. **Validate the microphysical parameterization** :

agree

3. **Validate the numerical model used and particularly the frost-dew deposition** :
I agree with your reply. Deposition (dew and fog settling) is clearly an important process in the formation phase of a fog layer, and I agree that it is an area in which NWP are deficient. Given the instrumentation deployed during LANFEX, you could perhaps discuss this point and discuss the impact of your modification on water deposition on ground. I think that the total water deposition on the ground could be more useful than the evolution of the specific humidity at screeen level.

For the soil-atmosphere exchanges, It would be nice to discuss the limitations of the appraoch used (imposed surface temperature and consequently no interaction between land and atmosphere). During the formation and dissipation phase, it seems that the surface - atmosphere interactions have a huge impact on fog life cycle. Therefore, your approach could be limiting.

4. **Contribution of this study with respect to bibliography** :
Agree fog Bott (1991). Your work should be discussed with respect to this reference.

The comparison of your results with the results of Maronga and Bosveld (2017) should be added in the revised version. I agree with your reply but this point should be clarified in the revised version.

---

## Author Comment (AC2) · 29 Jan 2018

**Aerosol-fog interaction and the transition to well-mixed radiation fog:**
**2nd Response to reviewer 1**

Ian Boutle, Jeremy Price, Innocent Kudzotsa, Harri Kokkola, Sami Romakkaniemi

January 29, 2018

We thank the reviewer for their follow up to our previous response - being able to discuss things like this is a huge benefit of the ACP format. Responses to the follow up are included below, along with a description of how the manuscript has been altered to address both sets of comments – any page or line numbers refer to the latexdiff file.

**1. Demonstrate the usefulness of using 3 types of models (LES - NWP - climate model) : I totally agree with you about complementarity between observations and LES. LES are a great tool to better understand the processes occuring during the fog life cycle**

We have tried to clarify the roles of each type of model in the introduction (P2, L15-19), making it clear that our focus is on evaluating and improving the NWP model, but we are utilising the LES as a process model to understand the observations. We have also added more discussion on the usefulness of the LES model (P3, L8-17), particularly to comment on how we are not reliant on a traditional parametrization of aerosol activation.

**I also agree that the ultimate goal is to improve NWP simulations, and a statistical study could demonstrate that this goal is achieved. In my opinion, the statistical validation of NWP is very interesting and needs to be included in the revised version. However, for fog (rare event) I am not sure that probability of false detection (b/(b + d)) would be the best indicator of false alarm because d $\gg$ b. I prefer the false alarm rate : b/((a + b)**

The statistical validation has now been included in the paper (Fig 12, P14 L10 - P15, L9). You have also spotted a typo / inability on my part to read the documentation - what we present as probability of false detection is indeed the false alarm rate b/(a+b).

**I am not convinced by the usefulness of climate simulations. In my opinion, this**

**part of the article makes the manuscript more confusing without added scientific values.**

We agree that the climate simulations are distinct from the rest of the paper. This is why they are included at the end of the conclusions section, essentially as a corollary to the rest of the paper, rather than part of the main text. However, we feel that it is an important issue, which has not been raised anywhere else in the literature, but does have important consequences for the climate modelling community. We have now included a reference to Vautard *et al.* (2009) (P17, L8-10) to further motivate why this is so important. As such, we would like to retain them in the paper as motivation that future work is needed on this topic.

**For the LES, it would be useful to study in detail the variability found in LES simulations, and to validate it with observations (if available). Moreover, what is the impact of microphysics in this variability?**

We agree that studying the variability in the LES would be useful. The observations taken as part of LANFEX were not targeted at the scales of LES variability - stations were located several kilometres apart, e.g. in adjacent valleys or on hilltop locations, to investigate the mesoscale variability in fog formation and evolution. Therefore studying the variability in our single LES would be somewhat unconstrained and distinct from the main focus of this paper. With that in mind, we are planning an LES intercomparison based around this IOP1, in which the variability and effects of microphysics on this will be a much greater focus. Therefore this is left as future work.

**I also have questions about activation processes in LES model. Given the time step used in LES study, I am not sure that a direct coupling between microphysics and LES updrafts (turbulent updrafts) is the best way to modelize the activation process. What is the representative time for activation processes? Is it compatible with the time step used in LES or with the lifetime of turbulent updrafts?**

The timestep of the LES is limited to a maximum of 0.5s, and in practice is dynamically shortened to between 0.2 and 0.3s after the turbulence within the fog has formed. Thus the timestep is clearly shorter than the time needed to simulate cloud activation, which occurs on the order of a few seconds depending on the updraft velocity or cooling rate. As far as we know, the direct coupling between the turbulence and microphysics is currently the only way to accurately simulate the activation, and also the evaporation of droplets within fogs or cloud. Typically used parametrizations for droplet formation are targeted at calculating activation at the cloud base and those are not valid in conditions with pre-existing droplets. With direct coupling we are also able to take into account the kinetic limitations related to condensation, i.e. particles are not always able to exceed their critical size and activate into droplets even though the ambient supersaturation might be higher than the particle critical

supersaturation for a short period of time. We have included a brief comment on this in the revised manuscript (P3, L24-26).

**2. Validate the microphysical parameterization: agree**

We have included a summary of the previous discussion in the revised manuscript (P3, L8-17).

**3. Validate the numerical model used and particularly the frost-dew deposition: I agree with your reply. Deposition (dew and fog settling) is clearly an important process in the formation phase of a fog layer, and I agree that it is an area in which NWP are deficient. Given the instrumentation deployed during LANFEX, you could perhaps discuss this point and discuss the impact of your modification on water deposition on ground. I think that the total water deposition on the ground could be more useful than the evolution of the specific humidity at screeen level.**

We have included the water deposition rate figure in the revised manuscript (Fig 4a), and the previous discussion about dew deposition rates and the results of Guedalia and Bergot (1994) (P6, L29 - P7, L6). This actually re-enforces our argument about the water contents being correct and the droplet numbers being the primary source of error, so thank you for this suggestion.

**For the soil-atmosphere exchanges, It would be nice to discuss the limitations of the appraoch used (imposed surface temperature and consequently no interaction between land and atmosphere). During the formation and dissipation phase, it seems that the surface - atmosphere interactions have a huge impact on fog life cycle. Therefore, your approach could be limiting.**

Yes, we agree that surface-atmosphere interaction is relevant for the life cycle both in terms of formation and dissipation. The goal of the LES exercise presented is not to study how environmental conditions are affecting the fog, which has been done in detail previously (e.g. Bergot *et al.*, 2015; Maronga and Bosveld, 2017). Here we have used the LES only to understand the coupling between aerosol and droplet activation, interaction with radiation and fog dynamics in the observed conditions. We have added a comment to the revised manuscript (P3, L19-22) to discuss that this may be a limitation.

**4. Contribution of this study with respect to bibliography : Agree fog Bott (1991). Your work should be discussed with respect to this reference.**

We have now placed the work of Bott (1991) up-front in the introduction as clear motivation for our own work (P2, L5-11, 15-16). We have also now stated that points (i)–(iii) are

a summary of the mechanisms affecting the development of well-mixed fog (P16, L11), not necessarily our own contribution.

**The comparison of your results with the results of Maronga and Bosveld (2017) should be added in the revised version. I agree with your reply but this point should be clarified in the revised version.**

We have included a paragraph on the comparison of our results to Maronga and Bosveld (2017) in the conclusions (P16, L20-29).

**References**

Bergot, T., Escobar, J., and Masson, V. (2015). Effect of small-scale surface heterogeneities and buildings on radiation fog: Large-eddy simulation study at parischarles de gaulle airport. *Q. J. R. Meteorol. Soc.*, **141**, 285–298.

Bott, A. (1991). On the influence of the physico-chemical properties of aerosols on the life cycle of radiation fogs. *Boundary-Layer Meteorol.*, **56**, 1–31.

Guedalia, D. and Bergot, T. (1994). Numerical Forecasting of Radiation Fog. Part II: A Comparison of Model Simulation with Several Observed Fog Events. *Mon. Weather Rev.*, **122**, 1231–1246.

Maronga, B. and Bosveld, F. (2017). Key parameters for the life cycle of nocturnal radiation fog: a comprehensive large-eddy simulation study. *Q. J. R. Meteorol. Soc.*, **143**, 2463–2480.

Vautard, R., Yiou, P., and van Oldenborgh, G.-J. (2009). Decline of fog, mist and haze in europe over the past 30 years. *Nat. Geosci.*, **2**, 115–119.

---

## Author Comment (AC3) · 29 Jan 2018

**Aerosol-fog interaction and the transition to well-mixed radiation fog:**
**Response to reviewer 2**

Ian Boutle, Jeremy Price, Innocent Kudzotsa, Harri Kokkola, Sami Romakkaniemi

January 29, 2018

We thank the reviewer for their positive and constructive comments. We have described below how the manuscript has been altered to address them – any page or line numbers refer to the latexdiff file.

**Abstract: -"Improvements to the representation of cloud droplet concentration" - it is more correct to say that you implemented a reduction of the concentration.**

This was already addressed before the discussion phase. We have modified the sentence as "Modifications to the parametrization of cloud droplet numbers in fog, resulting in lower and more realistic concentrations". Simply stating that we have reduced the concentration makes it sound like arbitrary tuning, which it is not - a key focus of the paper is demonstrating that drop numbers in fog are low and the parametrization systematically over-estimates them. Hopefully by expanding the sentence we have made it both clearer and more correct.

**Structure: In general the paper is well written, though I recommend the paper structure should be a bit more clearly presented in the beginning of the paper. The paper contains a discussion of observations, LES, and NWP model and a climate model. Some of these appearances appear a bit as a surprise or are only very briefly introduced, which give the reader a bit the uncertain feeling like "where do we go?". Hence the authors should better introduce why the chain of models presented in necessary to answer the research questions. Just some more details would be appreciated, which would also help to ensure reproducibility.**

We have endeavoured to address this comment in the revised manuscript. The aims are now more clearly stated in the introduction (P2, L15-16 - to understand the observations and evaluate/improve an NWP model). The methodology to achieve this is then presented through the sections (P2, L16-19), including the introduction of the LES as a process model to supplement the observations. We have also expanded the motivation and description of the LES

(P3, L5-26), making it clear how it differs from the NWP model as a tool for understanding the physical processes at work.

**Figures: I suggest to plot observations in dots and model results as lines so they are more easy to distinguish without reading the caption twice.**

This was already addressed before the discussion phase. We're not sure exactly which figures the reviewer is referring to, as many plots contain multiple observations, which are plotted in a manner suiting their measurement - lines to represent continuous or high frequency data, dots to show discrete measurements. To enable the reader to get a quick overview of the plots without having to read the caption, we have added the primary observation on each panel to the legend (typically a black line).

**Synthesis: I encourage the authors to strengthen the discussion section. Somehow I have the feeling the paper has a bit the nature of a technical report that present the impact of changes in model settings, that are in itself of course valuable, but the synthesis how the current results relate to other studies could be strengthened. At least I am aware of other studies that report on a much smaller sensitivity on the microphysical settings than presented here. As such a more complete synthesis would be valuable.**

We have altered both the introduction and conclusions in response to this comment. We now motivate the work more strongly (P2, L5-11) as a follow on to the work of Bott (1991), a paper which first describes the strong effect of aerosol on the fog life-cycle. We have then included a discussion in the conclusions to the recent paper of Maronga and Bosveld (2017), describing how although their results may appear quite different, we actually feel they are quite complimentary (P16, L21-29). We have also clarified a further point on the treatment of longwave scattering in NWP models, which may make our results different to others (P16, L30 - P17, L2).

**References**

Bott, A. (1991). On the influence of the physico-chemical properties of aerosols on the life cycle of radiation fogs. *Boundary-Layer Meteorol.*, **56**, 1–31.

Maronga, B. and Bosveld, F. (2017). Key parameters for the life cycle of nocturnal radiation fog: a comprehensive large-eddy simulation study. *Q. J. R. Meteorol. Soc.*, **143**, 2463–2480.

---

## Author Response (AR2)

**Response to referee 1**

Ian Boutle, Jeremy Price, Innocent Kudzotsa, Harri Kokkola, Sami Romakkaniemi

May 4, 2018

**The meticulous corrections of the authors have resulted in a manuscript that is greatly improved.**

Thank you!

**Yet I am left with an uncomfortable feeling about this paper. This work demonstrates that modifications in the droplet number could have a significant effect during the formation of fog. However the authors do not discuss in depth the physical mechanisms responsible of this sensitivity.**

This is a very good point! Reducing the cloud droplet number has two key effects on the fog development. Firstly, it increases the bulk fog droplet sedimentation rate, which is calculated (Wilkinson *et al.*, 2013) via Stokes Law as

$$\overline{V_c} = \frac{1.339 \times 10^6}{F(T)} \left( \frac{\rho q_c}{N_c} \right)^{\frac{2}{3}}, \tag{1}$$

where $F(T)$ is a temperature dependant function, $\rho$ is the air density, $q_c$ is the fog water content and $N_c$ is the fog droplet number. This increases the rate at which condensed water is removed from the fog layer onto the surface, physically thinning the fog layer, i.e. the liquid water path (LWP) is reduced. This is shown in Figure 4, where in the "Drop Taper" experiment, the LWP is approximately halved, yet the water deposition rate (which is almost entirely droplet settling) is largely unchanged.

Secondly, it reduces the cloud absorptivity to upwelling longwave radiation from the surface, which is calculated (Slingo and Schrecker, 1982) as

$$k(e) = q_c(a + b/r_e) \tag{2}$$

where $r_e$ is the effective radius and $a$ and $b$ are constants. Reducing the droplet number increases the effective radius, and so reduces the cloud absorption, allowing more radiation to be emitted to space. This reduces the effectiveness of the fundamental process by which radiation fog develops – longwave emission from the fog itself (which is proportional to the absorption). Cooling from the fog top is less efficient at turbulence production, and downwelling longwave radiation from the fog is less efficient at heating the surface (and thus enabling surface driven

turbulence to form). This is effect is demonstrated by the "No Rad Cld" experiment in the paper, in which $k(e)$ is set to 0 and fog development is significantly reduced. This discussion has been added to the paper at P13,L26-P14,L3.

**As said in conclusions, it seems that this sensitivity only appears for low values of drop numbers : Maronga and Bosveld (2017) and Steeneveld and Bode (2018) have found no sensitivity for values between 100 a 300 cm-3, and the authors found a strong sensitivity for values between 75 and 50 cm-3. This threshold sensitivity should be explained in details before the manuscript could be accepted for publication in ACP.**

The threshold sensitivity arises at the point at which the fog becomes optically thick (i.e. upwelling and downwelling longwave radiation balance) and turbulently well-mixed. Once this state is reached, further changes to the cloud droplet number do little to alter the evolution of the fog. Because the simulations of Maronga and Bosveld (2017) and Steeneveld and de Bode (2018) become instantly optically thick and well-mixed, and their chosen perturbations never push it out of this state, they do not find much sensitivity. It's only when the model can simulate fog in an optically thin (i.e. upwelling and downwelling longwave radiation not in balance) and turbulently stable state, that the droplet number provides a much bigger control as it can determine when the transition between these states occurs. It is this state which the modelling of this paper has explored, although the observations presented in all three studies show it to be happening. This discussion has been clarified in the paper at P17,L10-16.

**The first concern is the fact that the authors have omitted a thorough discussion of the microphysical processes related to the droplet number modification. What is the main impact of the modification of the fog droplets number on microphysical processes? effect on sedimentation? radiative effect? Why?**

Please see discussion above.

**Please quantify the effect of the microphysical modifications. Comparison with observation made during LANFEX (e.g. validation of the sedimentation with the measasured surface water deposition) could be helpful to better understand the threshold sensitivity found by the authors.**

The droplet sedimentation is shown in Figure 4, and discussion of this has been included (P13,L28-30) as part of the above discussion on the effects of droplet number on sedimentation rates. The downwelling longwave radiation (a reasonable proxy for cloud absorption) is shown in Figure 6, and discussion of this has been included (P13,L33) as part of the above discussion on the effects of droplet number on the radiation balance.

**Figure 5 shows that the simulated droplet number is about 5 times smaller than observed ones in the LES. Why? This seems to demonstrate that others microphysical processes are not correctly taken into account.**

The appropriate comparison for droplet number between the LES and observations is the dashed line, which samples all droplets greater than $2\mu$m in diameter, as the cloud droplet probe does. These values are well within the observational range shown by the instrument, and so we believe that the LES is a faithful simulation of reality. This is discussed in the text at P11,L20-P12,L8.

**The second concerns is the fact that some simulated parameters show significant errors with respect to observations near the surface before the adiabatic fog, except for LES where the temperature at the surface is forced : e.g sensible heat flux (fig2), water deposition rate (fig4),downwelling lonwave radiation (fig6). This seems to demonstrate that the coupling between the land-surface and atmosphere is not correctly simulated.**

We are unsure to what the reviewer is referring here. In the "Drop Taper" experiment, the sensible heat flux begins to diverge from the observations around 01 UTC, but even here the error is very small ($\approx 5$Wm$^{-2}$) and still a significant improvement on the "Control" simulation. It is likely this is because the droplet number is still too high at this point in the night – the observations and LES show it to be closer to 25cm$^{-3}$ in Figure 5 – and separate tests with a single column version of the UM (not shown) confirm this to be the case. But our aim is not to produce the best simulation possible for this case – that would require a parametrization of droplet number which varies throughout the night – it is to understand the processes at work, which we believe the simulation is good enough to do. The choice of a droplet number of 50cm$^{-3}$ is by no means perfect, but is a ballpark number representative enough of typical fog conditions and easy enough to implement in an NWP model.

The water deposition rate shows good agreement with the observations throughout the main foggy period (21 UTC – 5 UTC). It is deficient before 21 UTC, and this is thought to be because of hygroscopic absorption, a process not represented by this (or any) model. Also, during this early period, there is a very shallow and inhomogeneous fog present in reality, but it is too shallow to be simulated by the model (it's only several meters deep, and so below the lowest model level). This will also enhance the observed deposition rate. This is noted in the text at P7,L1-4.

The downwelling longwave radiation shows very good agreement with the observations, both in terms of the timing and magnitude of the increase from clear sky values around 21 UTC. The error in the clear sky value before this is most likely due to errors in the temperature and humidity of the upper troposphere, and perhaps some high-level cirrus cloud, but does not have a significant effect on the simulation.

In summary, we believe the interaction between the atmosphere and land surface is very well simulated, and the plots and analysis in the paper clearly justify this belief.

**Moreover, the evolution of observed profiles between 0000UTC and 0030UTC seems to demonstrate that processes at the fog top play a key role in the onset of adiabatic fog (local advection? KH waves?).**

Our belief, based on witnessing the event happening and knowledge of other fog events at the Cardington site, is that there is no evolution or development of the fog layer between 0000 and 0030 UTC - this is just natural variability in the fog top height and the luck of where the point measurement samples the fog. The thin fog is very inhomogeneous, whilst the LES and NWP model are presenting averaged quantities over a $\approx$ 2 km grid-square. The fog top naturally varies in height due to gravity waves, but this process is largely a reversible one and not responsible for development of the fog layer (the fluxes caused by the waves are negligible). This case was chosen for analysis because it presented one of the most clear cut examples of local/in-situ fog development we observed during LANFEX, and this conclusion is supported by the fact that we can produce a credible simulation of the event with no advective forcing of the LES model. Indeed we have tested with a single column version of the UM that there is no significant difference in the fog evolution with or without advective forcing. We have added some comments about this (P2,L26-28, P7,L11-12), although feel it largely detracts from the point that the NWP model is clearly wrong (has an adiabatic temperature profile) long before this time, and advective processes are not responsible for this error.

**The simulated temperature at 100m of LES also showed significant errors at 100m with respect to tethered balloon (fig3, fig10). Why? This seems to demonstrate that processes at the top of the fog layer are not correctly simulated.**

This is due to the decision to run the LES unforced (except for at the surface). The free troposphere cools by approximately 2K (the error shown in Fig 3 and 10) between 17 UTC when it is initialised and 22 UTC (the next radiosonde released), after which it remains approximately constant throughout the night. As mentioned above, including this forcing in a single column version of the UM makes no significant difference to the results, hence we feel it is an unimportant complication and choose to run the LES unforced.

**Given these remarks, the proposed modifications look like a way to compensate model misrepresentation of dynamical processes. These modifications are efficient for UKV (fig12). But is it specific to this model (due to errors in other parameterisations)?**

We do not believe that the proposed modifications are a compensation for mis-represented dynamical processes, which we do not believe were a significant feature in the development

of this fog event. We do believe that these modifications are genuine improvements to the simulation of radiation fog, and should be applicable to any model. However, without testing other models this is impossible to answer. For that reason, we are setting up a model inter-comparison project, initially based around this case, to better understand the sensitivities and behaviours of different NWP models. We have noted this in the text at P17,L17-20.

[revised manuscript text omitted]